# Synchrony between midbrain gene transcription and dopamine terminal regulation is modulated by chronic alcohol drinking

Zahra Z. Farahbakhsh[1], Katherine M. Holleran[2], Jonathon P. Sens[2], Steve C. Fordahl[3], Madelyn I. Mauterer[2], Alberto J. López[1], Verginia C. Cuzon Carlson[4], Drew D. Kiraly[2], Kathleen A. Grant[4], Sara R. Jones[2] & Cody A. Siciliano[1] ✉

Alcohol use disorder is marked by disrupted behavioral and emotional states which persist into abstinence. The enduring synaptic alterations that remain despite the absence of alcohol are of interest for interventions to prevent relapse. Here, 28 male rhesus macaques underwent over 20 months of alcohol drinking interspersed with three 30-day forced abstinence periods. After the last abstinence period, we paired direct sub-second dopamine monitoring via ex vivo voltammetry in nucleus accumbens core with RNA-sequencing of the ventral tegmental area. We found persistent augmentation of dopamine transporter function, kappa opioid receptor sensitivity, and putative dynorphin release – all inhibitory regulators which act to decrease extracellular dopamine. Surprisingly, though transcript expression was not altered, the relationship between gene expression and functional readouts of these encoded proteins was highly dynamic and altered by drinking history. These results outline the long-lasting synaptic impact of alcohol use and suggest that assessment of transcript-function relationships is critical for the rational design of precision therapeutics.

Alcohol Use Disorder (AUD) is defined by numerous symptoms involving over-prioritization and -dedication to alcohol use, including craving, excessive consumption, and continued use in the face of negative consequences[1]. Unfortunately, AUD is a chronic relapsing disorder, and even with treatment, relapse rates after achieving abstinence remain as high as 40–60%[2,3]. Decades of clinical and preclinical research has shown that protracted abstinence from chronic drinking is characterized by depression-like behaviors, increased response to alcohol-related cues, and greater stress-induced craving,

making periods of abstinence a critical intervention point to treat individuals susceptible to relapse[4–10]. Leading models of AUD neurobiology posit that complex, alcohol-induced changes in gene expression and translation result in time-dependent, neural circuit-specific functional plasticity, which in turn drive the cardinal behavioral repertories of AUD[11–16]. Central to these models, the mesolimbic dopamine circuit, consisting of dopamine neurons projecting from the ventral tegmental area (VTA) to the nucleus accumbens core (NAc), undergoes dramatic functional plasticity in response to chronic

---

[1]Department of Pharmacology, Vanderbilt Brain Institute, Vanderbilt Center for Addiction Research, Vanderbilt University, Nashville, TN 37232, USA. [2]Wake Forest University School of Medicine, Department of Physiology and Pharmacology, Winston-Salem, NC 27157, USA. [3]The University of North Carolina at Greensboro, The Department of Nutrition, Greensboro, NC 27412, USA. [4]Oregon National Primate Research Center, Oregon Health & Science University, Division of Neuroscience, Portland, OR, USA. ✉e-mail: cody.siciliano@vanderbilt.edu

alcohol exposure and there is wide consensus that this circuit plays an essential role in many facets of AUD[17–19]. Leveraging current detailed knowledge of alcohol-induced transcriptional plasticity and circuit function has enormous potential for developing more effective, targeted therapeutic interventions, and this possibility has captured the attention and efforts of the preclinical alcohol field at large.

Although alcohol-induced plasticity in the mesolimbic dopamine system is robust, the directionality and magnitude of functional alterations to the receptors and transporters governing dopamine dynamics are highly dependent on the exact parameters of alcohol exposure and the model species used, despite consistent behavioral sequalae[20,21]. Cross-species comparisons have allowed identification of a subset of dopaminergic regulators which display consistent alcohol-induced adaptations. These studies have demonstrated that the efficiency of the dopamine transporter and inhibitory drive of kappa-opioid receptors on presynaptic dopamine release are ubiquitously upregulated by alcohol exposure across species, strain, sex, alcohol exposure length, and alcohol exposure protocol[21–30]. The consistency of these alterations makes them attractive treatment targets, due to the likelihood of these changes being conserved in human drinkers and given that most therapeutics are administered without the benefit of detailed information as to the patient's exact alcohol use history. However, it is unknown whether these alterations persist into abstinence.

Additionally, to date, our knowledge of alcohol-induced transcriptional and functional plasticity signatures is derived primarily from parallel investigations, and both literatures have largely utilized non-primate model species with limited gene homology to humans. Determining the degree to which transcriptional milieus can predict circuit plasticity profiles and advancing these hypotheses to primate species are critical next steps for the field. Here, we sought to determine whether alcohol-induced plasticity in dopamine reuptake and opioidergic control of axonal dopamine release persist into abstinence, and the degree to which lasting plasticity in dopamine synapses can be explained by upstream changes in somatic transcriptional profiles.

To answer these questions, we paired a model of chronic voluntary alcohol (ethanol) drinking and protracted abstinence with within-subject measures of dopamine terminal function and assessment of gene transcription. In an effort to maximize translational relevance and implications of gene expression quantification, rhesus macaque (*Macaca mulatta)* subjects were used, given that they display behavioral diversity which mirrors individual differences in human drinkers[31], as well as homology to humans in genetic sequence[32] and neural circuit architecture[33,34]. Following an induction protocol, subjects were allowed to self-administer alcohol under continuous access conditions for 12 months, followed by three one-month abstinence periods interspersed with three-month periods of re-access to alcohol. At the end of the third and final abstinence period, real-time dopamine release kinetics and inhibitory regulation by axonal kappa opioid receptors were measured directly via ex vivo fast-scan cyclic voltammetry in the NAc core (17 alcohol drinkers, 11 calorically yoked or housing controls). The findings of the voltammetry experiments included 1) that dopamine transporter and kappa opioid receptor plasticity persisted into protracted abstinence, 2) that changes in evoked dopamine release magnitude did not persist, except at the extremes of the input/output curve, 3) that G protein signaling was both necessary and sufficient for kappa opioid receptor inhibitory control of axonal dopamine release but did not account for alcohol-induced supersensitivity, 4) that kappa opioid receptor co-activation was necessary for corticotropin-releasing factor (CRF) receptor-mediated inhibition of dopamine release but was not altered by drinking history, and 5) that dynorphin release probability, measured using a time-resolved readout, was increased in protracted abstinence.

To probe mechanisms that may underlny functional differences observed by fast-scan cyclic voltammetry, bulk RNA-sequencing of the VTA was conducted to measure gene expression upstream of dopamine terminals. Given the involvement of the mesolimbic system in addiction-related behaviors, we hypothesized that chronic alcohol consumption and abstinence would result in robust changes in VTA gene expression, and that plasticity in dopamine terminal function would be related to changes in the expression of dopamine-specific genes. Contrary to our hypothesis, we found zero differentially expressed genes in the VTA between drinkers and controls and minimal evidence for any network-level dysregulation. Unexpectedly, within-subject correlations between fast-scan cyclic voltammetry measures, such as release magnitude and rate of reuptake, and the expression of genes encoding for proteins thought to be functionally involved, such as D2-type autoreceptors and dopamine transporter, showed that VTA transcript expression and NAc function are often decorrelated in controls. Further, relationships between expression and function are experience-dependent, as drinking resulted in the emergence or change in directionality of many transcript × function relationships. These findings directly call into question the common assumption that transcriptional expression is positively correlated with functional outputs of the encoded proteins – instead, we show that the existence and directionality of these relationships are highly plastic and cannot be assumed. Together, this work probes the complex and dynamic relationships between function and gene expression and demonstrates multiple plasticity mechanisms by which chronic alcohol consumption and abstinence modulate the mesolimbic dopamine circuit.

## Results
### Subjects displayed wide individual difference in alcohol consumption
With the goal of translationally modeling human alcohol consumption and relapse, a drinking protocol was devised such that intake was voluntary, alcohol was available for prolonged periods, and macaques had access to alcohol restored after periods of forced abstinence[31,35–40]. Briefly, 17 alcohol drinking rhesus macaques underwent a schedule-induced polydipsia procedure to initiate voluntary alcohol consumption, followed by 12 months of 22 hours/day open access drinking, a month of forced abstinence, 3 months of open access alcohol reintroduction, a month of forced abstinence, 3 months of open access alcohol reintroduction, and one month of forced abstinence at the end of which subjects were necropsied (Fig. 1A). Through this protocol, individual differences emerged in average alcohol intakes (g/kg) during the first open access period (Fig. 1B), and both reintroduction periods (Fig. 1C), as well as the sum total of lifetime alcohol consumption (Fig. 1D). Blood was collected longitudinally over the course of alcohol access periods and assayed for blood alcohol concentration, which strongly correlated with the subject's alcohol consumption during the collection day (Fig. 1E). Experiments were performed in two cohorts of subjects which each contained controls and drinkers, and did not differ in alcohol consumption levels (see Methods and Fig. S1A–D), body weight, or age (Fig. S1E, F). Immediately following necropsy, the brain was blocked into sections including the NAc and the VTA[41]. The NAc was kept under physiological conditions for live ex vivo slice recordings whereby fast-scan cyclic voltammetry was used to measure sub-second dopamine release dynamics and probe multiple aspects of dopamine terminal function. The VTA, taken from the same subjects, was flash frozen and subsequently bulk RNA-sequencing was used to probe upstream alcohol-induced changes in transcription. Altogether, this design allowed for the investigation of how chronic drinking and protracted abstinence alter presynaptic dopamine terminal function in the NAc core as well as transcriptional changes in the VTA. A unique feature of this within-subject design is the ability to assess relationships between endogenous gene

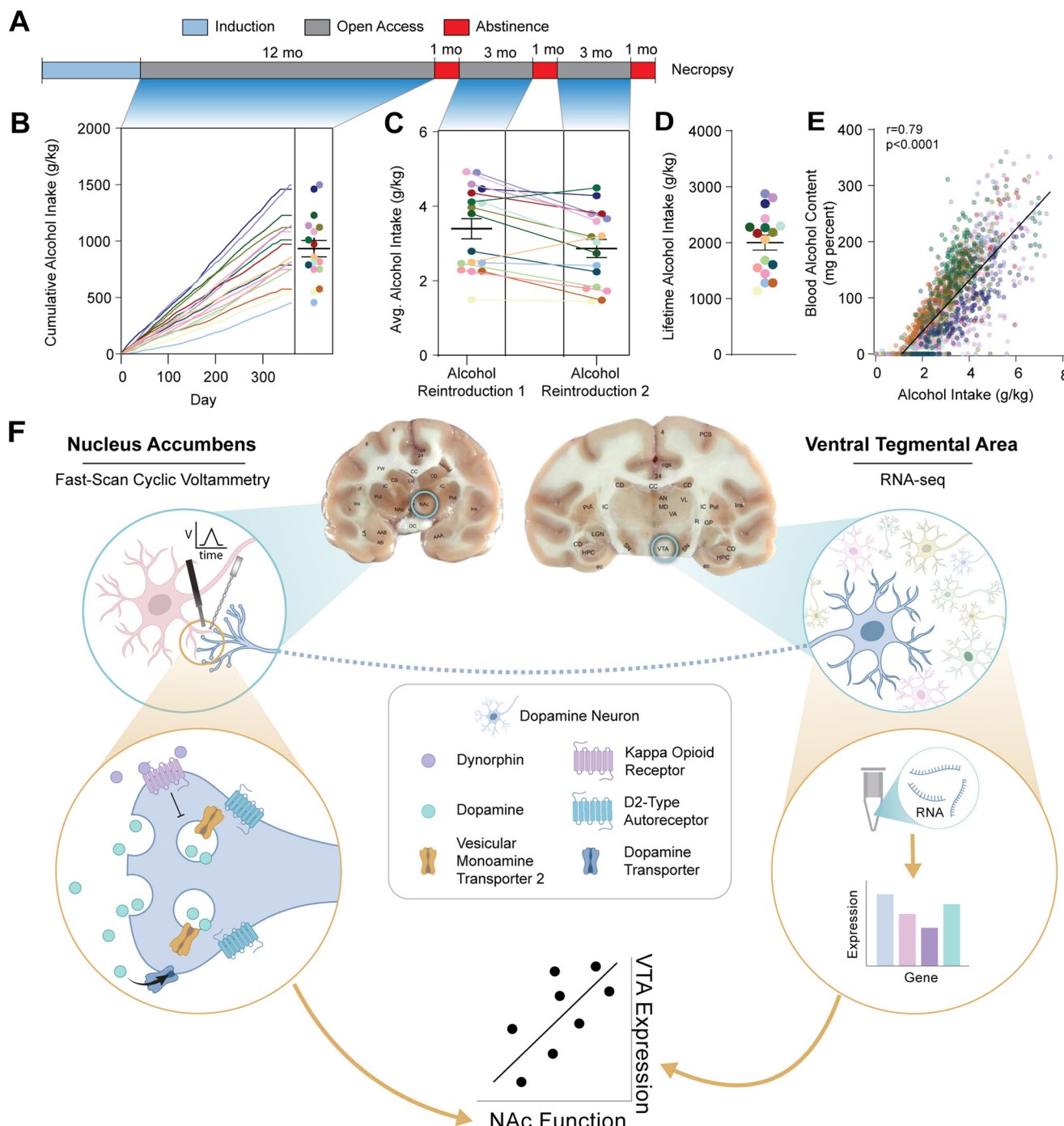

**Fig. 1 | Diagram of experimental design and rationale. A** Rhesus macaques (17 alcohol and 11 calorically yoked or housing controls) underwent a drinking (or housing control) protocol designed to uncover individual differences in drinking phenotypes between subjects. Briefly, a schedule induced polydipsia procedure was used to induce voluntary alcohol consumption, followed by 12 months of 22 hour/day open access drinking, a month of forced abstinence, 3 months of open access alcohol reintroduction, a month of forced abstinence, 3 months of open access alcohol reintroduction, and one month of forced abstinence at the end of which subjects were necropsied. **B–E** Drinking data from each epoch of the self-administration paradigm is shown, with color indicating the same subject throughout. **B** Cumulative alcohol intake was calculated over the first period of open access. **C** Average daily alcohol intake during each of the two alcohol reintroduction periods. **D** Lifetime intake in g/kg was calculated for each of the 17 alcohol-exposed subjects. **E** Blood alcohol concentration (in milligram percent [i.e. mg/dL]) was collected weekly, 7 hours after session start, for each subject and was strongly positively associated with the alcohol intake on the same day (two-tailed Pearson's correlation). **F** After necropsy, the brain was blocked in coronal sections including the nucleus accumbens (NAc) core and the ventral tegmental area (VTA); dopamine dynamics were recorded from the NAc with fast-scan cyclic voltammetry, and gene expression from the upstream VTA region was assessed via bulk RNA-seq. Stimulation parameters and pharmacological manipulations were used to assess different features of dopamine terminal release in the NAc. These effects were then correlated with gene expression measures from the VTA to assess the relationship between terminal function and upstream transcription. Unless otherwise indicated, values indicate mean ± SEM. (drinkers: $n = 17$) Created in BioRender. Siciliano, C. (2025) https://BioRender.com/l92m987.

expression in the midbrain soma and functional measurement of axonal dopamine release, reuptake, and regulation (Fig. 1F).

## Protracted abstinence from chronic drinking results in minimal transcriptional perturbation in the VTA

To search for potential mechanisms mediating long-term neuronal dysregulation induced by cycles of chronic alcohol consumption and abstinence, full transcriptomic RNA-sequencing was conducted on VTA tissue, which houses the somatic compartment of the mesolimbic dopamine circuit, following the alcohol self-administration procedure described above. Given the persistence of aberrant behaviors associated with heavy drinking, and the large body of work implicating the mesolimbic dopamine system in alcohol-related behaviors, we hypothesized that transcriptional networks, and dopamine neuron-specific genes in particular, would be markedly dysregulated following abstinence from chronic drinking. In stark contrast to our hypothesis, differential expression analysis showed that, following Benjamini-Hochberg false discovery rate adjustment, zero genes were differentially expressed between drinkers and controls (Fig. 2A). Also contrary to our hypothesis, but corroborating the lack of differential gene expression, dimensionality reduction via principal component analysis did not produce appreciable segregation between drinkers and controls across pairwise visualizations of the top five principal components (Figs. 2B, S2). This suggests that long periods of abstinence were not characterized by gross changes in the expression of individual genes in the VTA.

To evaluate whether transcriptional differences were maintained within networks of co-expressed genes, a weighted gene correlation network analysis was conducted[42] (Fig. 2C, D). This analysis allows for the clustering of co-expressed genes into network modules, without a priori information of gene function[43] (Supplemental Dataset 1). Through this analysis, we found 4 modules that were correlated with the control condition, and one module correlated with having a history of alcohol self-administration (Fig. 2E). Using gene ontology analysis, the module most correlated with controls, *dark grey*, was enriched for pathways including vesicle-mediated transport, such as the kinesin-like protein, KIF1C, and ATPase activity, such as probable ATP-dependent RNA helicase 10, DDX10 (Fig. 2F). The module *sienna*, most correlated with alcohol drinkers, was enriched for genes involved in the postsynapse, such as semaphorin 6 A (sema6a) and misfolded protein binding, among others (Fig. 2G). From this data, we concluded that there were minimal global changes in gene expression within the VTA that were induced by repeated cycles of chronic alcohol drinking and long-term abstinence and thus, behavioral effects of abstinence are likely maintained through mechanisms beyond up- or downregulation of specific genes or networks.

## Alcohol consumption constrains dynamic range of accumbal dopamine release and augments reuptake rate

In parallel, we sought to assess if functional alterations in accumbal dopamine terminals were present in these subjects. Voltammetric detection of extracellular dopamine concentration was used to monitor evoked release and reuptake kinetics in real-time in ex vivo brain slices[44]. We first assessed the excitability of dopamine terminals by comparing release magnitude across a range of stimulation intensities (50-900 µA, single pulse, 4 ms, monophasic stimulations) (Fig. 3A). We found that, while groups displayed similar $EA_{50}$ values (Excitatory Amplitude 50, or the stimulation amperage required to produce a half-maximal excitatory dopamine response), the relationship between stimulation intensity and dopamine release was altered by alcohol history such that drinkers showed decreased dopamine release evoked by high intensity stimulations. Further, chronic drinking and abstinence resulted in decreased dynamic range of dopamine release, as measured by the differential between the upper and lower

plateaus of the input-output curve (Fig. 3B–D). For the remainder of experiments, release was evoked by 350 µA stimulations, near the shared $EA_{50}$ for drinkers and controls (Fig. 3E).

Evidence suggests that ongoing alcohol intake produces a hypo-dopaminergic state through increased activity of inhibitory pre-synaptic regulators, including increased rate of dopamine reuptake from the extracellular space through augmented DAT efficiency[21,22,25,26,28–30], but whether this effect is maintained throughout protracted abstinence from voluntary alcohol drinking remains untested. One pulse stimulations (350 µA, 4 ms, monophasic) were used to evoke dopamine release (Fig. 4A). Observed peak dopamine concentrations are a product of the amount of dopamine released and the rate of ongoing dopamine reuptake, parameters which can be dissociated using Michaelis-Menten modeling. Using this approach, we found no group difference in dopamine release evoked by a 350 µA stimulation (Fig. 4B). However, maximal rate of dopamine reuptake ($V_{max}$) was increased in subjects with a history of chronic alcohol self-administration (Fig. 4C). These results are consistent with a hypodo-paminergic state[45–47], which is consistently observed in the mesolimbic system of humans with AUD and thought to be critical to symptomatology[48–52], and provide evidence that this state persists in prolonged abstinence.

## Chronic drinking induces intercompartmental synchrony between somatic transcription and axonal dopamine dynamics

We next asked how these persistent changes in dopamine terminal function were maintained through abstinence, considering that the abstinence period is 5-10 fold longer than the biological half-life/turnover rate of the dopamine transporter protein (3-6 days)[53,54], and it can therefore be reasonably assumed that none of the dopamine transporters present in the brain at this point have ever interacted directly with alcohol or its metabolites. To address this question, we assessed the within-subject relationship between dopamine terminal function in the NAc and the upstream expression of genes known to encode for regulators of dopamine release and reuptake localized on accumbal dopamine terminals arising from the VTA. Thus, dopamine release (Fig. 4D–G) and $V_{max}$ (Fig. 4H–K) were correlated with the VTA expression of a priori selected genes encoding for: the D2-type dopamine autoreceptor (*DRD2*; ENSMMUG00000014334), the kappa opioid receptor (*OPRK1*; ENSMMUG00000065247), the dopamine transporter (*SLC6A3*; ENSMMUG00000005198; referred to throughout by the gene alias *DAT*), and the vesicular monoamine transporter-2 (*SLC18A2*; ENSMMUG00000014817; referred to throughout by the gene alias *VMAT2*). The receptors encoded by *DRD2* and *OPRK1* are $G_{i/o}$ coupled GPCRs which are highly expressed on presynaptic dopamine terminals in the NAc core, and activation of either receptor results in decreased dopamine release and dopamine terminal excitability[55–57]. The expression of both *DRD2* and *OPRK1* in the VTA was surprisingly not correlated with dopamine release in the NAc in controls, but showed a positive correlation in drinkers (Fig. 4D–E). *DRD2* expression was also not correlated with $V_{max}$ in controls but showed a trend toward a positive correlation in drinkers (Fig. 4H) and neither group showed a relationship between $V_{max}$ and *OPRK1* expression (Fig. 4I). The transporters encoded by *DAT* and *VMAT2* are enriched in dopamine-releasing neurons in the VTA and are responsible for the reuptake through the cytoplasmic membrane and the vesicular packaging of dopamine, respectively (Fig. 1E)[47,58,59]. Once again, only in drinkers, expression of the *DAT* and *VMAT2* transporter encoding genes in the VTA showed a trend toward a positive correlation with dopamine release (Fig. 4F, G) and were positively correlated with $V_{max}$ (Fig. 4J, K). Thus, across these a priori genes of interest, variance stabilized read counts from the VTA and dopamine dynamics in the NAc did not correlate in control subjects, contrary to the canonical assumption that transcript expression is tightly related to protein expression and thereby function.

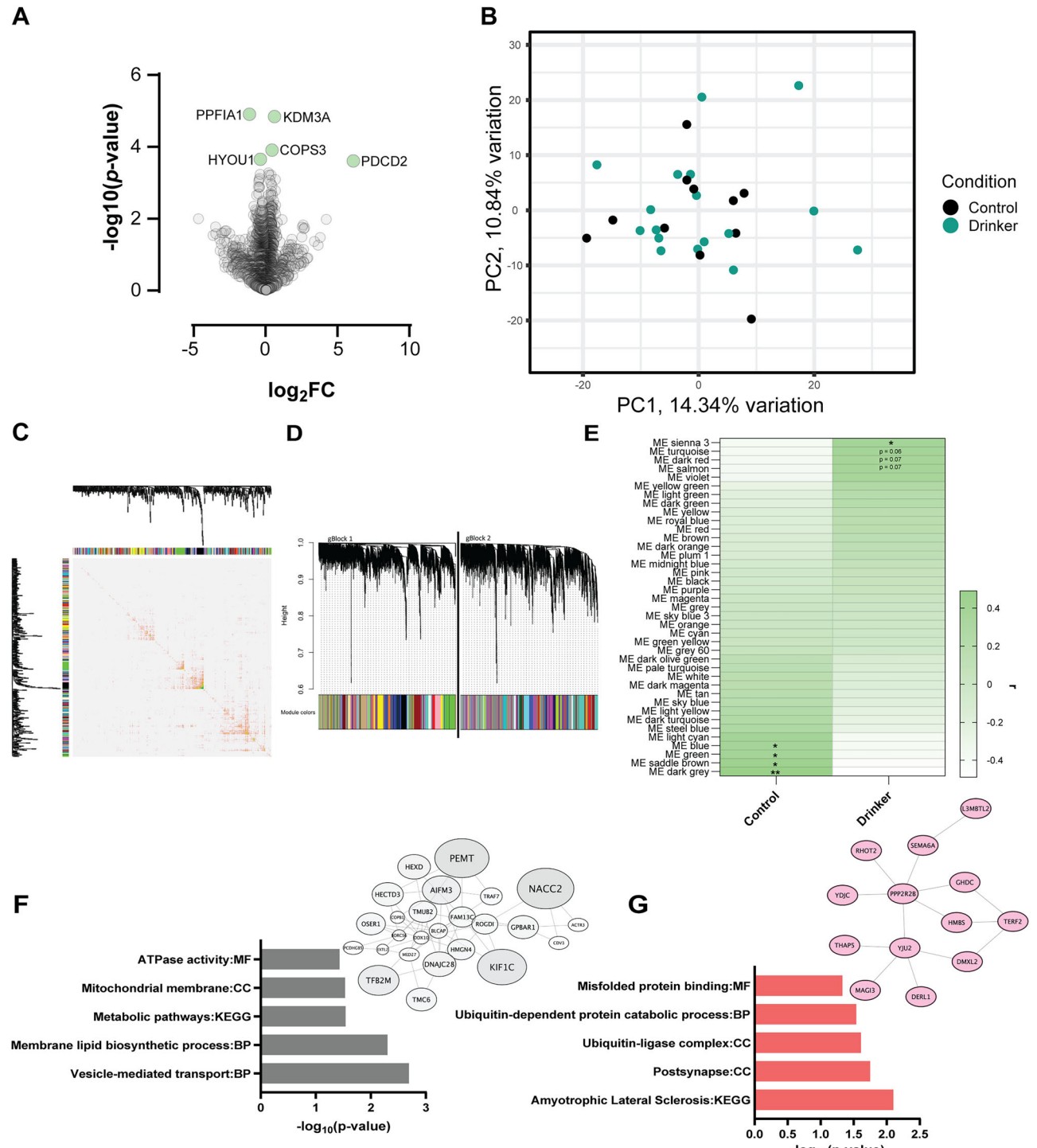

**Fig. 2 | Minimal changes to the expression of individual genes or coexpression networks in midbrain following chronic alcohol intake and protracted abstinence.** Deep sequencing was performed on the ventral tegmental area (VTA) from 28 macaques (paired-end 150 bp, roughly 150 million read pairs per sample). Reads were aligned to the *Macaca mulatta* genome (Mmul_10) and read counts were calculated and normalized for each subject. **A** A volcano plot of gene expression showing log₂(fold change) between drinkers and controls and the raw p-value. After Benjamini Hochberg FDR <0.05 correction of p-values, zero genes were significantly different between drinkers and controls. While no genes passed false-discovery rate correction, the transcripts with the five lowest $p$ values are labeled and highlighted in green. **B** Dimensionality reduction via principal component

analysis did not produce appreciable segregation between drinkers and controls across pairwise visualizations of the top 2 principal components. **C** Adjacency plot of weighted co-expression between the top 500 genes defining modules after WGNA analysis. **D** Tree dendrogram of weighted gene co-expression. Module eigengenes that reached a threshold of 0.99 or above were included for network analysis. **E** Heatmap showing module-trait relationships by group from WGCNA analysis. Color axis indicates Pearson's $r$ value. **F** Genes and connectivity, and significant GO terms of interest of the *dark grey* module most correlated with controls. **G** Genes and connectivity, and significant GO terms of interest of the *sienna 3* module most correlated with alcohol drinkers. All statistical tests were two-tailed. (*$p \leq 0.05$, **$p \leq 0.01$, ***$p \leq 0.001$, ****$p \leq 0.0001$) (controls: $n = 11$; drinkers: $n = 17$).

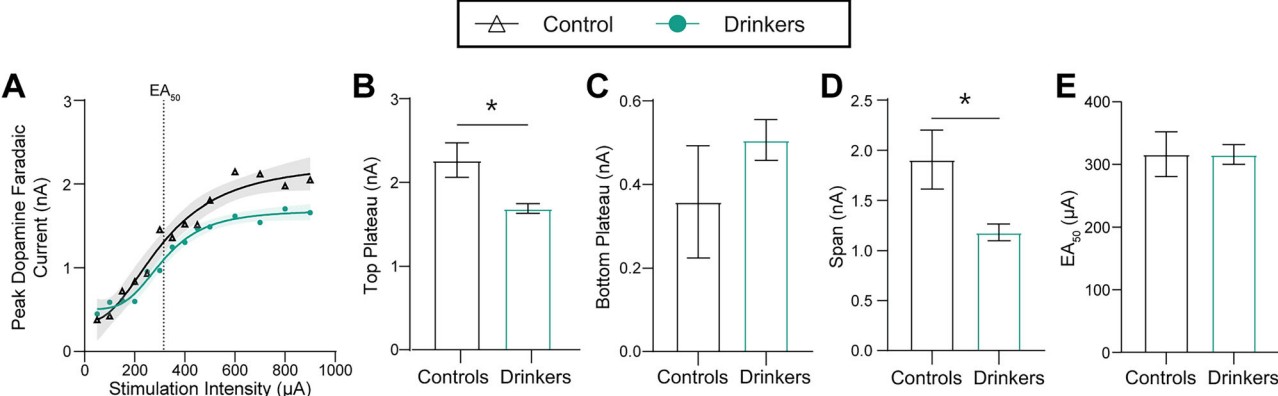

**Fig. 3 | Chronic voluntary alcohol consumption retricts dynamic range of accumbal dopamine release. A** An input-output curve showing the peak dopamine release (nA) evoked by single pulse stimulations across ascending intensities (50−900 µA). Curves were fit with a 4-parameter sigmoidal regression, and best-fit values are shown with 95% confidence band and half-maximal excitatory amperage (EA$_{50}$) indicated (controls: EA$_{50}$ = 316.4 µA; drinkers: EA$_{50}$ = 315.9 µA). **B**−**D** Comparison of best-fit values between drinkers and controls. **B** Subjects with ethanol history have an attenuated upper asymptote compared to controls, indicative of decreased maximal dopamine release magnitude (unpaired *t*-test;

$t_{26}$ = 2.710, *p* = 0.0118). **C** There is no difference between groups at the bottom of the curve plateau (unpaired *t*-test; $t_{26}$ = 1.033, *p* = 0.3110). **D** Drinkers show a decreased span of the input-output curve [upper plateau minus lower plateau] compared to control subjects suggesting a lower dynamic range of dopamine release (unpaired *t*-test; $t_{26}$ = 2.365, *p* = 0.0258). **E** There was no difference in the EA$_{50}$ between drinkers and controls (unpaired *t*-test; $t_{26}$ = 0.01, *p* = 0.99). (controls: *n* = 3; drinkers: *n* = 7) All statistical tests were two-tailed and values indicate mean ± SEM. (**p* ≤ 0.05, ***p* ≤ 0.01, ****p* ≤ 0.001, *****p* ≤ 0.0001).

The a priori aim of the analyses described above was to assess how increased transcriptional expression might be driving persistent changes in dopamine reuptake throughout abstinence. Unexpectedly, these analyses revealed a striking pattern which was apparent across comparisons whereby the expression of individual genes appeared decorrelated with dopamine release and reuptake in controls, but displayed synchrony between expression and function in animals with a history of alcohol use. Indeed, a comparison of the slopes of the linear regressions for each group revealed that a history of alcohol intake and abstinence increased the mean slope compared to controls (Fig. 4L). Further, comparison within each group revealed that drinkers showed a positive mean slope, whereas the mean slope for controls did not differ from zero (Fig. 4L). This pattern raises the intriguing possibility that cycles of alcohol use and abstinence may alter the fundamental relationship between somatic transcriptional activity and axonal function independent of changes, or lack thereof, in transcript expression levels. To explore this possibility, data were transformed to a within-group rank order for each dependent variable. Thus, each subject was assigned a value from 1 to 8 denoting lowest to highest expression for each of the 4 transcripts, paired with a 1 to 8 value indicating rank order of V$_{max}$ and a 1 to 8 value indicating rank order of dopamine release magnitude. This allowed all 8 correlations (64 total x-y pairs per group) to be plotted and analyzed in a single coordinate plane with rank-expression on the y-axis and rank-function on the x-axis. Overall, there was no discernable relationship between gene expression and function in controls, with an *r*-value approaching zero (Fig. 4M). On the other hand, there was a strong positive relationship, indicating tight synchrony between gene expression and terminal function, after chronic alcohol consumption and abstinence (Fig. 4N).

Given the surprising, experience-dependent induction of inter-compartmental synchrony between transcription and function, and the striking degree of similarity across the correlations in drinkers, we next sought to probe the veracity of these findings and test whether these relationships showed selectivity for the genes of interest. Thus, we next performed a secondary analysis on a set of genes selected *a posteriori*. This set was matched in size (4 transcripts) and type of encoded protein (2 GPCR encoding genes, and 2 solute carrier-family transporter encoding genes), but were selected based on known enrichment in non-dopamine releasing neurons within the VTA[58]. Expression of the genes encoding for dopamine

receptor 1 (*DRD1*; ENSMMUG00000005147), dopamine receptor 5 (*DRD5*; ENSMMUG00000002783), vesicular glutamate transporter 2 (*SLC17A6*; ENSMMUG00000020368; referred to throughout by the gene alias *VGLUT2*), and vesicular GABA transporter (*SLC32A1*; ENSMMUG00000003376; referred to throughout by the gene alias *VGAT*) was thus compared with dopamine release and reuptake using the same analyses as described above (Fig. S3A-H).

Importantly, we found that transcriptional expression of these *a posteriori*-selected genes within the VTA were poor predictors of downstream dopaminergic function within the NAc in both controls and drinkers, corroborating the relationships observed in drinkers for dopamine-regulating transcripts are unlikely to be epiphenomenological. Across the *a posteriori*-selected gene set, there was only one positive correlation found: that between *DRD5* expression and dopamine release in controls (Fig. S3B) that was lost in drinkers. Further, the mean slope of the linear regressions did not differ from zero in both controls and drinkers (Fig. S3I). Likewise, when gene expression and terminal function were transformed to a within-group rank order, as described above, there was no correlation between rank expression and rank function in drinkers or controls (Fig. S3J-K). In the dopamine-regulating transcripts, the apparent induction of synchrony between transcription and function occurred in the absence of differential expression of these transcripts between groups (Fig. S4). Of interest, neither dopamine release, dopamine reuptake, nor transcriptional expression levels correlated with lifetime alcohol intake in the drinkers (Figs. S5, S6). Together, these data show that a history of alcohol consumption induces synchrony between the transcriptional activity of autoregulatory dopaminergic modulators during protracted abstinence. Broadly, these results also call into question long-held assumptions regarding transcriptional control of synaptic function.

## Supersensitivity of axonal kappa opioid receptors in the accumbens persists into protracted abstinence

One of the inhibitory regulators of terminal dopamine release, the kappa opioid receptor, has been shown to play a causal role in behaviors characteristic of AUD and may serve as a potential therapeutic target. A large body of work across species and methods of alcohol administration has shown that sensitivity of the kappa opioid receptor is augmented by chronic, ongoing alcohol exposure, resulting in hyperpolarization of dopamine terminals and thereby likely

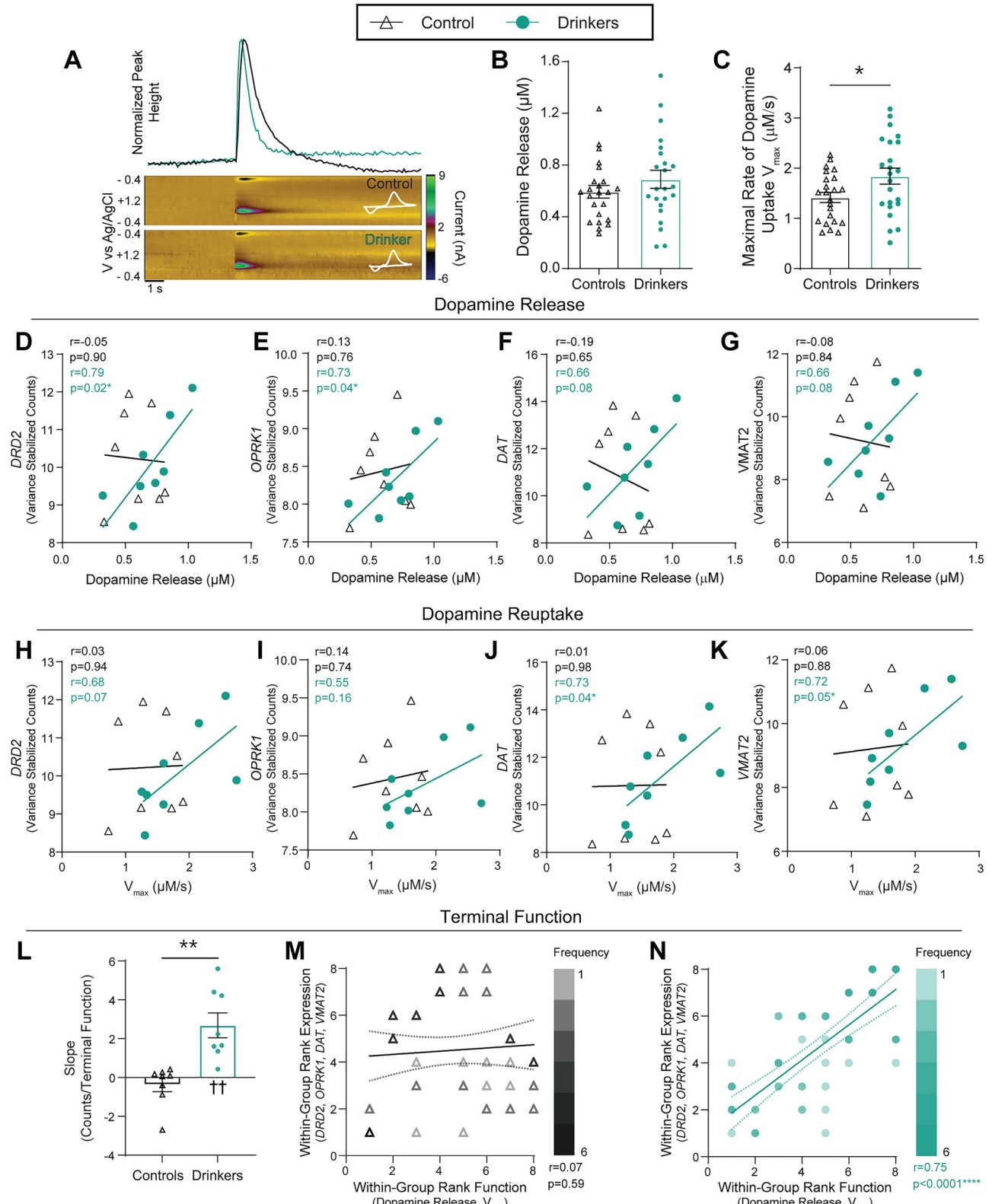

contributing to the hypodopaminergic state seen in AUD[24,27–30]. The importance of alcohol-induced plasticity of the kappa opioid receptor system is underscored by findings highlighting a role for selective kappa opioid receptor antagonists in the treatment of AUD[28,60–65]. Despite the apparent importance of kappa opioid receptor activity in ongoing drinking behaviors, the precise signaling mechanisms underlying kappa opioid receptor inhibition of dopamine release remains unclear, and whether drinking-induced plasticity in this system persists beyond the acute withdrawal phase remains untested.

To functionally assess the sensitivity of kappa opioid receptor regulation of presynaptic dopamine release, a selective, unbiased kappa opioid receptor agonist, U50,488 was bath applied to ex vivo slices and 1 pulse dopamine release was recorded and compared to pre-drug baseline values (Fig. 5A). Dopamine release was inhibited to a

**Fig. 4 | Chronic drinking induced long-lasting changes in dopamine reuptake concomitant with synchrony between upstream transcription and downstream dopamine release dynamics. A** Representative normalized concentration versus time traces, pseudo-color plots, and current by voltage traces for a drinker and a control subject. **B** There was no difference in dopamine release between drinkers and controls (unpaired $t$-test; $t_{44} = 1.13$, $p = 0.2647$). **C** Alcohol history increased the rate of dopamine reuptake ($V_{max}$) (unpaired $t$-test; $t_{44} = 2.239$, $p = 0.0302$). **D–K** The best-fit linear regression is plotted and Pearson's correlation coefficient $r$ and $p$ values are reported as an inset. Covariance between expression and function emerged following drinking: $r$ values of 0.6 or greater were observed for (**D**) the dopamine receptor 2 (*DRD2*), (**E**) the kappa opioid receptor (*OPRK1*), (**F**) the dopamine transporter (*DAT*), (**G**) and the vesicular monoamine transporter 2 (*VMAT2*). **H** For *DRD2*, there was a trend towards a positive correlation between gene expression and $V_{max}$ in drinkers, not controls.

**I** Expression of *OPRK1* was not correlated with $V_{max}$ in either group. $V_{max}$ and transporter expression did not correlate in controls, but in drinkers, $V_{max}$ showed a positive correlation with the expression of *DAT* (**K**) and *VMAT2* (**J**). **L** The mean slope of gene expression over terminal function was greater in drinkers (unpaired $t$-test; $t_{14} = 4.14$, $p = 0.001$). The mean slope differed from zero in drinkers (one sample $t$-test; $t_7 = 4.18$. $p = 0.004$), but not in controls (one sample $t$-test; $t_7 = 1.01$, $p = 0.35$). **M, N** The best-fit linear regression is shown with a 95% confidence band. Inset: Spearman's $r$- and $p$-values. **M** There was no correlation between expression of homosynaptic regulators and terminal release in control subjects. **N** Alcohol history induced a positive correlation between gene expression and function. All statistical tests were two-tailed and unless otherwise indicated, values indicate mean ± SEM. (*$p \leq 0.05$, **$p \leq 0.01$, ***$p \leq 0.001$, ****$p \leq 0.0001$, ††$p \leq 0.01$ vs 0) (controls: $n = 8$ [23 slices]; drinkers: $n = 8$ [23 slices]).

greater degree across concentrations of U50,488 in drinkers compared to controls, demonstrating that supersensitivity of the kappa opioid receptor persists through long-term periods of abstinence (Fig. 5B). In separate slices, a non-selective inhibitor, 500 μM barium chloride (BaCl$_2$)[66], was used to block G protein-coupled inwardly rectifying K$^+$ channels (GIRKs)—a downstream effector activated by kappa opioid receptors among others[67,68], which act to reduce membrane excitability. Bath application of BaCl$_2$ induced a greater increase in evoked dopamine release in drinkers compared to controls, suggesting augmented basal GPCR-mediated inhibitory regulation of dopamine terminals in the NAc core (Fig. 5C).

### G protein activity is necessary and sufficient for kappa opioid receptor inhibition of axonal dopamine release, but does not account for alcohol-induced supersensitivity

We next sought to determine which pathway was involved in the supersensitization of the receptor seen with a history of alcohol intake. Many agonists of the kappa opioid receptor are biased toward either the G protein or β-arrestin signaling cascades downstream of the receptor[69,70]. Each of these pathways has been shown to drive differential cellular and behavioral outcomes of kappa opioid receptor activation[71–73]. Thus, we aimed to elucidate whether kappa opioid receptor control of dopamine release in the NAc was mediated via the G protein or the β-arrestin pathway. To achieve this goal, we utilized various pharmacological agents with specificity towards targets involved in the G protein or β-arrestin signaling cascades (Fig. S7A). An unbiased agonist, such as U50,488, activates both pathways to a similar degree and, in combination with inhibitors of the β-arrestin and G protein pathways, can be used to dissociate the necessity of each in the induction of kappa opioid receptor-mediated dopamine inhibition. Compound 101 (CMPD101) is a GRK2/3 inhibitor, which results in inhibition of β-arrestin signaling by preventing phosphorylation necessary for its recruitment. Thus, in the presence of CMPD101, application of U50,488 would result in preferential recruitment of effectors in the G protein pathway, thereby testing the necessity of the β-arrestin cascade[74]. N-ethylmaleimide (NEM), in contrast, is a sulfhydryl alkylating agent shown to inhibit G protein signaling, therefore in combination with U50,488, the β-arrestin pathway would be preferentially activated and G protein necessity revealed[75].

On separate slices from both drinkers and controls, either NEM (50 μM) or CMPD101 (30 μM) was bath applied and evoked dopamine was allowed to re-stabilize. Subsequently, U50,488 (1 μM) was added to the bath to determine agonist-induced effects on dopamine release in the presence of each inhibitor. When normalized to pre-drug baseline dopamine release, there was a differential effect of NEM and CMPD101 through consecutive wash-ons with U50,488 (Fig. S7B). When NEM was present, there was no change in dopamine release with the application of U50,488, suggesting that the G protein pathway is necessary for the effect, but there was a near significant decrease in dopamine release in the presence of CMPD101 (Fig. S7C). The ability of NEM to block kappa

opioid receptor inhibition of dopamine release supports the idea that kappa opioid receptor-mediated dopamine downregulation is mediated through G protein downstream signaling cascades.

Given the necessity of G protein signaling in kappa opioid receptor-mediated dopamine inhibition, we next wanted to determine whether this signaling cascade was sufficient to regulate release. To this end, nalfurafine, a biased kappa opioid receptor agonist which preferentially activates G protein signaling over the β-arrestin pathway[76], was used on separate slices. Indeed, nalfurafine was sufficient to inhibit release in both drinkers and controls; however, in contrast to U50,488, there were no group differences in the ability of nalfurafine to decrease release magnitude (Fig. 5D). Together, these pharmacological results suggest that G protein signaling may be crucial to kappa opioid receptor-mediated dopamine inhibition, but increased receptor sensitivity induced by chronic drinking may be due to alterations in other signaling mechanisms such as β-arrestin activity.

### Chronic drinking reverses the intercompartmental relationship between somatic transcription and axonal kappa opioid receptor regulation

To determine whether this altered function in the NAc core was associated with upstream VTA transcriptional changes, the magnitude of U50,488-induced decrease in dopamine release in the NAc was correlated with the expression of genes in the VTA encoding the kappa opioid receptor (*OPRK1*),the precursor to its primary endogenous ligand, prodynorphin (*PDYN*; ENSMMUG00000009984), β-arrestin 1 (*ARRB1*; ENSMMUG00000017529), and β-arrestin 2 (*ARRB2*; ENSMMUG00000017654). Both β-arrestins are important effectors of kappa opioid receptor, contributing to ligand-gated and constitutive activity[77–79]. The concentrations of U50,488 used, 300 nM, and 1 μM, allowed us to probe two facets of the kappa opioid receptor's control over dopamine release: sensitivity (moderate concentration, 300 nM), and efficacy (saturating concentration, 1 μM)[28]. When comparing *OPRK1* expression and the decrease in dopamine release at 300 nM U50,488, a linear association was only observed in drinkers, such that the greater the *OPRK1* expression in the VTA, the lower the sensitivity of the kappa opioid receptor (Fig. 5C). There was also a trend toward this negative correlation in drinkers with *PDYN* expression and potency of U50,488, respectively (Fig. 5D). There was no significant relationship between *ARRB1* or *ARRB2* expression in the VTA and the effect of U50,488 at 300 nM in the NAc, (Fig. 5G, H). At the high concentration (1 μM) of U50,488, *OPRK1* and *PDYN* showed a trend toward a positive correlation in controls alone (Fig. 5I, J). *ARRB1* expression in the VTA was not significantly correlated with kappa opioid receptor-mediated inhibition of accumbal dopamine release in drinkers or controls (Fig. 5K). However, in drinkers only, there was a correlation between the ability of the high dose of U50,488 to inhibit dopamine release and *ARRB2* expression, whereby greater inhibition was associated with lower transcriptional expression (Fig. 5L). Qualitatively similar to the alterations in expression × function

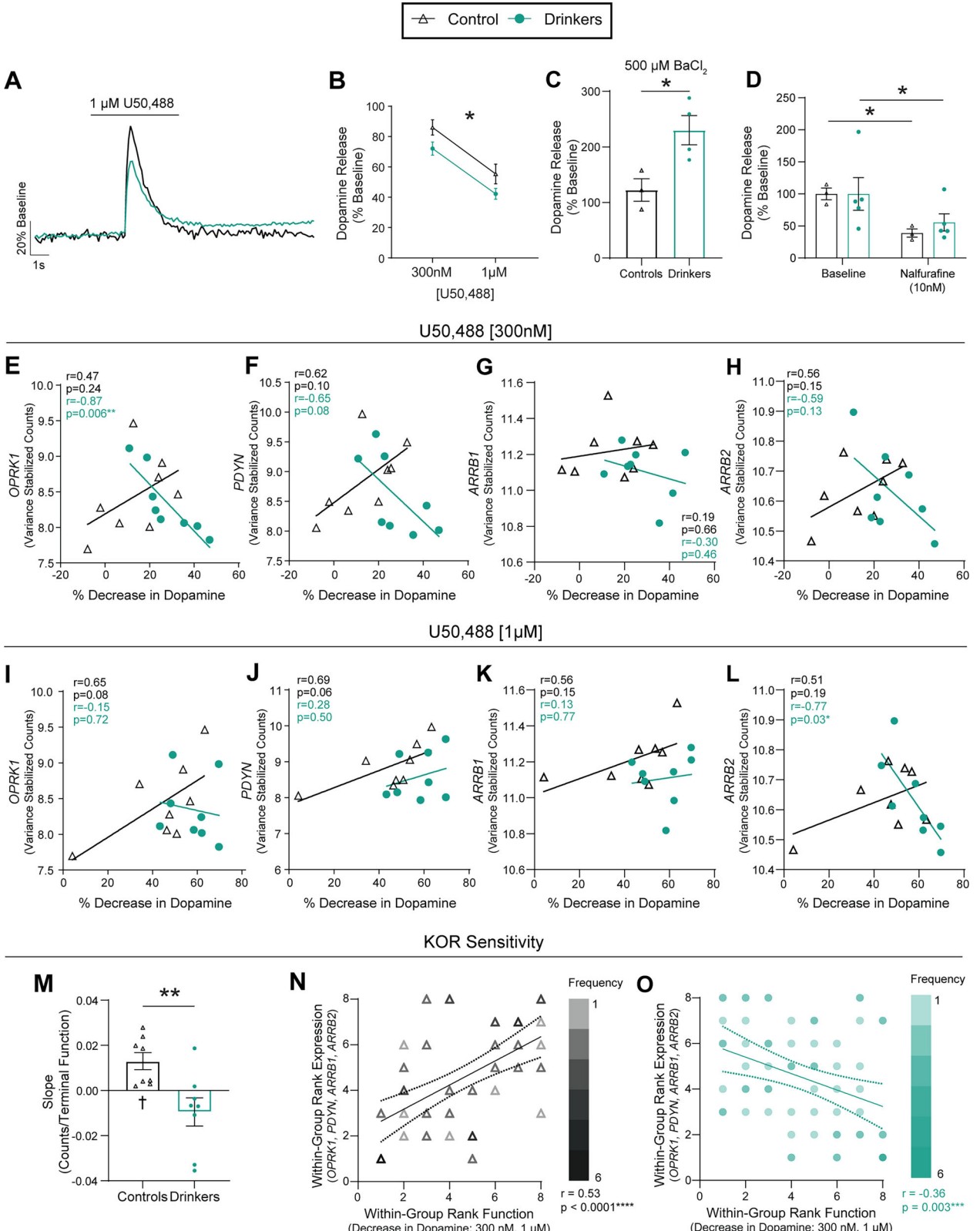

relationships for dopamine release dynamics reported above, the slope of the best-fit linear regressions between *OPRK1, PDYN, ARRB1,* and *ARRB2* expression in the VTA and kappa opioid receptor agonist-induced inhibition of accumbal dopamine release displayed an alcohol-induced sign change.

To probe the expression × function relationship pattern, we directly compared the slopes between drinkers and controls and

found that controls showed a positive mean slope that was greater than that seen in drinkers (Fig. 5M). Similar to the effect seen with homosynaptic regulators of dopamine release, this effect on the association between kappa opioid receptor-mediated regulation and expression of kappa opioid receptor-associated proteins supports the hypothesis that a history of alcohol intake and protracted abstinence modulates the relationship of transcriptional activity in the somatic

**Fig. 5 | Chronic drinking drove persistent upregulation of kappa opioid receptor control of dopamine release and altered function-transcription relationships. A** Representative dopamine release (1 μM U50,488). **B** U50,488 decreased dopamine release to a greater extent in drinkers (repeated measures two-way ANOVA; dose: $F_{1, 14} = 79.28$, $p < 0.0001$; group: $F_{1, 14} = 4.81$, $p < 0.05$; dose x group: $F_{1, 14} = 0.01$, $p = 0.91$). **C** $BaCl_2$ increased dopamine release to a greater extent in drinkers (unpaired $t$-test; $t_5 = 3.41$, $p = 0.02$; controls: $n = 3$, drinkers: $n = 4$). **D** Nalfurafine decreased dopamine release in both groups (two-way ANOVA; concentration: $F_{1, 6} = 30.76$, $p = 0.0015$; group: $F_{1, 6} = 0.10$, $p = 0.76$; concentration x group: $F_{1, 6} = 0.75$, $p = 0.42$; Šídák's test: baseline vs. nalfurafine; controls: $t_6 = 4.06$, $p = 0.01$, $n = 3$; drinkers: $t_6 = 3.82$, $p = 0.02$, $n = 5$). **E–O** Best-fit linear regression. Inset: Pearson's $r$ and $p$-values. **E** Change in dopamine release by U50,488 (300 nM) was correlated with *OPRK1* expression in drinkers, not

controls. **F** U50,488 potency was not correlated with *PDYN*, (**G**) *ARRB1*, or (**H**) *ARRB2* expression in either group. **I** There was no association between *OPRK1* expression and efficacy of 1 μM U50,488 in controls. **J** Efficacy was not associated with *PDYN* or (**K**) *ARRB1* in either group. **L** *ARRB2* and efficacy were not correlated in controls, but negatively correlated in drinkers. **M** Mean slope of the regressions was greater in controls (unpaired $t$-test; $t_{14} = 3.08$, $p = 0.008$). The mean slope for controls (one sample $t$-test; $t_7 = 3.45$, $p = 0.01$), but not drinkers (one sample $t$-test; $t_7 = 1.52$, $p = 0.17$), was greater than zero. **N** In controls, ranked expression was positively correlated with rank sensitivity. **O** This association was reversed in drinkers. All statistical tests were two-tailed. Unless otherwise indicated, values indicate mean ± SEM. (*$p \leq 0.05$, **$p \leq 0.01$, ***$p \leq 0.001$, ****$p \leq 0.0001$, †$p \leq 0.05$ vs 0) (unless otherwise noted, controls: $n = 8$; drinkers: $n = 8$).

compartment (VTA) with downstream functional outcomes (NAc core). To further probe the modulation of this relationship by alcohol history, dependent variables were again rank ordered. The lowest expression of *OPRK1*, *PDYN*, *ARRB1*, and *ARRB2*, and the lowest potency and efficacy of kappa opioid receptor control of dopamine release to the highest expression and greatest action of the receptor were assigned a value of 1 to 8, respectively. This allowed for all 8 of the correlations to be plotted and analyzed in a single coordinate plane with rank-expression on the y-axis and rank-function on the x-axis. In control subjects, there was a strong positive correlation between kappa opioid receptor transcript expression and the ability of U50,488 to inhibit dopamine release, suggesting synchrony between VTA transcription and functional sensitivity of axonal kappa opioid receptors in the NAc (Fig. 5N). A history of chronic alcohol self-administration and abstinence reverses this effect such that in drinkers, more VTA expression of *OPRK1*, *PDYN*, *ARRB1*, and *ARRB2* is correlated with less kappa opioid receptor control over accumbal dopamine release (Fig. 5O). Thus, voluntary alcohol intake and protracted abstinence can reverse the relationship between expression of genes encoding elements of the kappa opioid receptor system and its ability to inhibit dopamine release in the NAc.

These effects were once again independent of differential gene expression between drinkers and controls (Fig. S4B, S4I-K) and did not correlate with lifetime alcohol intake (Fig. S5B, S5I-K), demonstrating that alcohol self-administration, and abstinence in particular, modulates the relationship between transcription and function independent of overall transcript expression levels in the VTA. We also assessed whether the ratio of ligand to receptor (*PDYN/OPRK1*) expression in the VTA was correlated with functional measures in the NAc core, given the close relationship between endogenous ligand levels and receptor sensitivity and previous work showing alcohol experience-dependent relationships with this ratio[62]. There was no relationship between *PDYN/OPRK1* VTA expression and accumbal dopamine regulation in either the drinkers or controls (Fig. S8). Ultimately, this demonstrates that the supersensitivity of the kappa opioid receptor which is induced by chronic alcohol self-administration[21,27–30] persists at least one month into abstinence and is associated with an inversion of the relationship between expression of *OPRK1*, *PDYN*, *ARRB1*, and *ARRB2* and the ability of kappa opioid receptors to inhibit dopamine release.

**Kappa opioid receptor activation is necessary for CRF-mediated dopamine inhibition**
Given that responses to stressors are altered during abstinence and in light of evidence suggesting that CRF can affect the function of kappa opioid receptors[80,81], we also probed potential interactions between the CRF and kappa opioid receptor systems in regulating dopamine release in the NAc. Urocortin, an endogenous CRF receptor agonist, had no effect on dopamine release when bath applied alone in control and alcohol slices (Fig. S9A). However, the application of U50,488 in the presence of urocortin decreased dopamine release in both

controls and drinkers similarly (Fig. S9B). When urocortin was applied after U50,488, CRF receptor activation was able to further decrease dopamine release in both groups (Fig. S9C). Urocortin had no effect on dopamine release in the presence of a kappa opioid receptor antagonist, norbinaltorphimine (NorBNI) (Fig. S9D). Together, this suggests that kappa opioid receptor activation is necessary for CRF-mediated dopamine inhibition, and that activation of the CRF receptor is able to occlude the supersensitization of the kappa opioid receptor system seen at this abstinence timepoint.

**Alcohol history increases dynorphin release probability**
Prior work as well as the results presented up to this point have examined the effect of exogenous activation of the kappa opioid receptor system on dopamine release dynamics in the NAc. Here, we sought to stimulate the release of endogenous ligands of the kappa opioid receptor and infer the effects of kappa opioid receptor activation by examining dopamine release before and after application of a kappa opioid receptor antagonist. To measure endogenous activation of the kappa opioid receptor, the highly selective kappa opioid receptor antagonist NorBNI (10 nM) was used; any change in dopamine release magnitude due to the application of NorBNI is attributable to the blockade of endogenous activation of kappa opioid receptors, putatively by dynorphin, the primary known endogenous ligand of kappa opioid receptors. This pharmacological strategy was employed such that NorBNI-induced disinhibition of dopamine release evoked by multi-pulse stimulations was used as a proxy for dynorphin release probability. In both drinkers and controls, dopamine release was measured in response to 5 pulse stimulations delivered at 5 to 100 Hz to establish a pre-drug baseline, then these stimulations were repeated in the presence of the kappa opioid receptor antagonist NorBNI.

In controls, NorBNI did not modulate release magnitude across the stimulation frequency curve (Fig. 6A–C), indicating no measurable impact of endogenous dynorphin release evoked by the stimulations. In drinkers on the other hand, application of NorBNI produced a clear disinhibition of dopamine release (Fig. 6D). A comparison of release over stimulation frequencies before and after NorBNI application in drinkers showed a significant effect of NorBNI on dopamine release such that, predominantly at high frequency stimulations, NorBNI increased dopamine release (Fig. 6E). NorBNI also increased the area under the curve (AUC) of dopamine over stimulation intensities in subjects with a history of alcohol self-administration (Fig. 6F). The increase of dopamine release with NorBNI seen in drinkers suggests that kappa opioid receptors on terminals were endogenously activated by high frequency stimulations after chronic alcohol intake and protracted abstinence, most likely via increased release probability of dynorphin.

Interestingly, dynorphin release probability, quantified as the percent change in pre- and post-drug AUC for each group, was positively correlated with upstream *OPRK1* and *PDYN* RNA levels in controls, but not drinkers (Fig. 6G, H). These effects were once again not

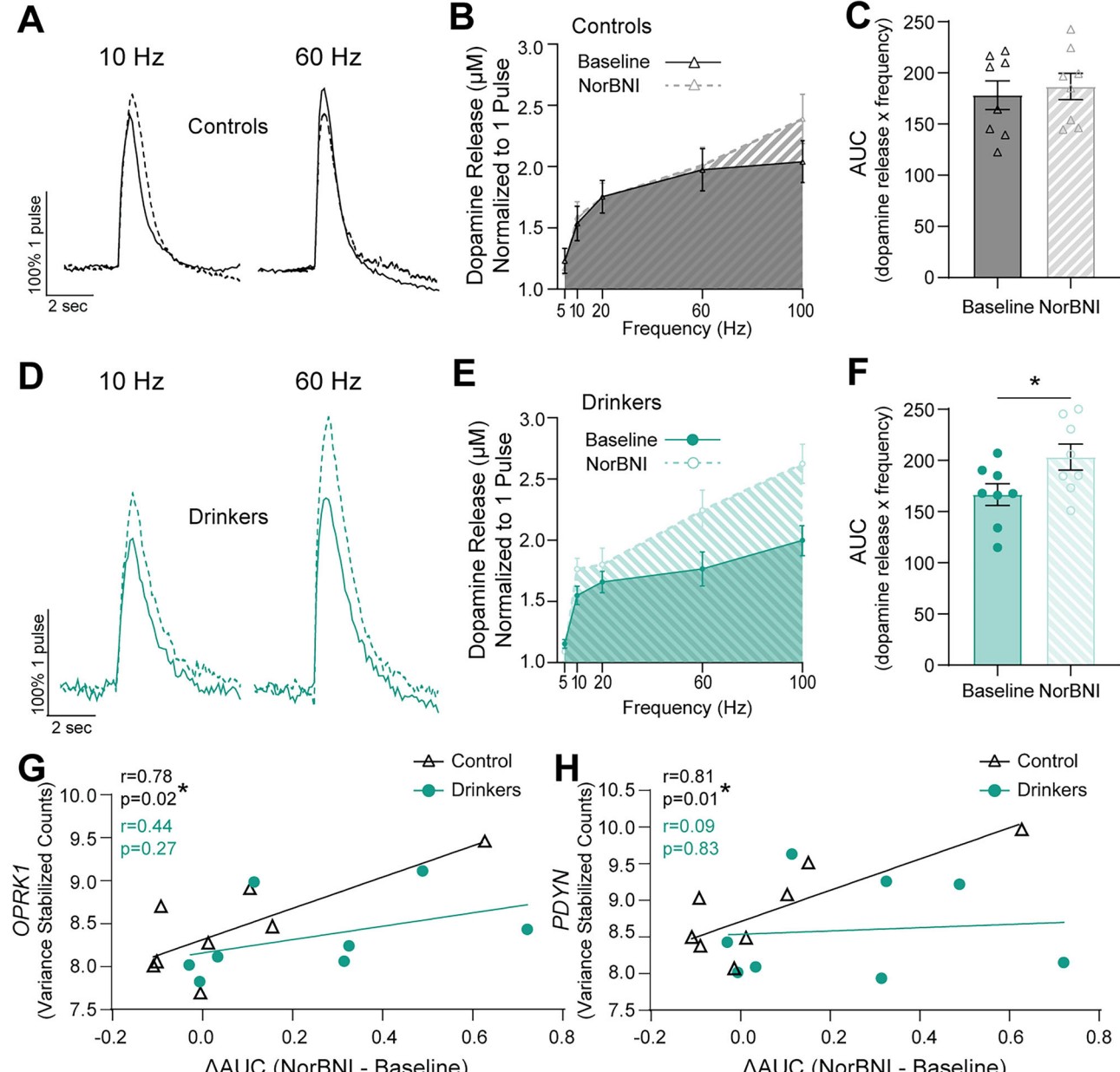

**Fig. 6 | Chronic alcohol consumption increased dynorphin release probability in long-term abstinence. A** Representative traces indicating dopamine release in controls at 10 Hz and 60 Hz normalized to 1 pulse dopamine release before and after NorBNI. **B** Dopamine release in controls with and without NorBNI across stimulation frequencies. **C** There was no effect of a history of ethanol intake, but there was an effect of NorBNI, on the area under the curve (AUC) of dopamine release across frequencies (mixed model two-way ANOVA: ethanol history: $F_{1,\,14} = 0.03$, $p = 0.88$; drug: $F_{1,\,14} = 6.152$, $p = 0.03$). NorBNI had no effect on AUC of dopamine release in controls (Šídák's test: baseline vs. NorBNI in controls: $t_{14} = 0.66$, $p = 0.77$). **D** Representative traces indicating dopamine release in controls at 10 Hz and 60 Hz normalized to 1 pulse dopamine release with and without NorBNI. **E** Dopamine release in drinkers with and without application of NorBNI across stimulation frequencies. AUC of dopamine release is visually represented by

figure shading. **F** NorBNI administration increased the AUC of normalized dopamine release in drinkers suggesting significant dynorphin release at these stimulation intensities after chronic alcohol self-administration (mixed model two-way ANOVA: ethanol history: $F_{1,\,14} = 0.03$, $p = 0.88$; drug: $F_{1,\,14} = 6.152$, $p = 0.03$; Šídák's test: baseline vs. NorBNI in drinkers: $t_{14} = 2.85$, $p = 0.03$). **G, H** Upstream expression of *OPRK1* and *PDYN* were correlated within-subject with the proportion of change in AUC of dopamine release. The best-fit linear regression is shown for each group. Inset: Pearson's $r$ and $p$-values. **G** Expression of *OPRK1* in the ventral tegmental area was positively correlated with the change in dopamine release with NorBNI administration in controls, not drinkers. **H** *PDYN* expression was also only correlated with the dynorphin release probability in controls, not drinkers. All statistical tests were two-tailed. Unless otherwise indicated, values indicate mean ± SEM. (*$p \le$ 0.05, **$p \le$ 0.01, ***$p \le$ 0.001, ****$p \le$ 0.0001) (controls: $n = 8$; drinkers: $n = 8$).

associated with previous lifetime alcohol intake and occurred in the absence of group differences in expression of these genes (Figs. S4, S5, S6). In sum, this suggests that chronic alcohol intake and long-term abstinence augments kappa opioid receptor control over dopamine release through the increase in release probability of dynorphin in the NAc and via a mechanism that disrupts the synchrony between functional release and upstream transcription.

## Discussion
The mesolimbic dopamine system undergoes dramatic neuroplasticity after chronic alcohol consumption and is a target for existing and promising therapeutics for AUD. A wealth of literature has suggested that the dopamine system is hypofunctional after chronic alcohol exposure, creating a hypodopaminergic state that is implicated in impaired decision-making associated with AUD. Here, we used RNA-

seq and fast-scan cyclic voltammetry to assess upstream transcription in the VTA and its relationship to downstream dopamine terminal function, respectively, at the end of a one-month abstinence period in drinkers with a history of alcohol intake (and calorically-yoked and housing controls). Though no genes were differentially expressed between drinkers and controls, and most expression networks were unchanged by a history of alcohol self-administration and repeated episodes of abstinence, clear functional changes at the dopamine terminal were seen after protracted abstinence. The maximal rate of dopamine reuptake, kappa opioid receptor sensitivity, and dynorphin release probability were upregulated in drinkers compared to controls, supporting the hypothesis of a hypodopaminergic state during abstinence. To determine potential mechanisms by which these long-lasting changes were maintained, measures of dopamine terminal function were correlated with VTA expression of genes encoding for regulators of these pathways. In the case of dopamine release dynamics, synchrony between function and transcription was only present in drinkers, while there was a reversal in the relationship between kappa opioid receptor/dynorphin transcription and receptor sensitivity in these subjects. Together, this work demonstrates that abstinence is characterized by persistent modulation of dopamine terminal activity that is associated with changes in the dynamic relationship between gene transcription and function. Importantly, these results suggest that, for any disorder, assessment of transcript-function relationships is critical for the rational design of gene-based precision therapeutics.

Clinical literature has shown that AUD is associated with marked hypofunction of dopaminergic activity in the ventral striatum which can be observed into abstinence[48,49,51,52]. Here we demonstrate that multiple inhibitory regulators of dopaminergic activity, dopamine transporters and kappa opioid receptors, are functionally upregulated in protracted abstinence from alcohol, providing a putative mechanism for the lasting hypodopaminergic state observed in AUD. Minimal work has assessed the persistence of plasticity at specific receptors and transporters into protracted abstinence and the molecular mechanisms that underly them. By demonstrating augmented dopamine reuptake, kappa opioid receptor sensitivity, and dynorphin release probability that exists after protracted abstinence, we highlight functional processes that are disrupted with chronic alcohol consumption and offer evidence supporting the potential utility of these pathways as therapeutic targets during the critical abstinence intervention point. Though more work needs to be done, the initial findings suggest that kappa opioid receptor control of dopamine release is G protein mediated while alcohol-induced upregulation may be related to altered β-arrestin. Together, this suggests that biased kappa opioid receptor ligands may be a beneficial approach to developing therapeutics for AUD and limiting off-target effects.

Initially, we hypothesized that plasticity that persisted into abstinence might be explained by alcohol-induced changes in gene expression. Surprisingly, there were no differentially expressed genes and few altered gene networks between subjects with a history of alcohol self-administration and controls. With an approximately 93% sequence identity homology between humans and rhesus macaques[32], the dataset generated from this work may still inform future investigations into the mechanisms underlying AUD and therapeutic targets. Importantly, there was no significant differential expression of genes encoding for proteins that make up the dopamine and kappa opioid receptor systems (e.g., VMAT2, DRD2, OPRK1). This is in line with previous post-mortem human work demonstrating no difference in overall expression of OPRK1 or PDYN in the nucleus accumbens of individuals with a history of AUD, but that AUD is associated with complex co-expression and transcriptional coordination of dopamine related genes[62]. When we correlated upstream transcription of these genes with downstream function of dopamine terminals, we found that, although they were not differentially expressed, transcription of

these genes of interest was closely associated with differential function. Understanding the complex relationship between transcription and function is critical to the interpretation of RNA-seq data which is often used to draw conclusions regarding circuit and synaptic activity. Typically, gene expression is assumed to be indicative of protein expression, despite reports that protein and transcript levels are often inversely correlated[82]. Previous work has also demonstrated that the ratio between RNA and protein levels is dependent on tissue type[83] or aging[84], but here, we go a step further by comparing transcript expression with measures of protein function and showed that this relationship, or lack thereof, is dynamic and experience-dependent. For example, the correlation between transcription and function, which is often assumed to exist, in some cases is not even present in controls but rather only induced with long term alcohol exposure and abstinence. For other genes and functions, a history of alcohol intake eliminates or even reverses the correlation between transcription and terminal activity that is seen in controls. Ultimately, altered transcription does not imply directionality or the presence of changes in function and vice versa; rather the relationship between transcriptional expression and function should not be stated or implied without quantitative assessments supporting any given claim. While the current dataset cannot speak to causality between the expression and function of any given transcript, we show that across dopamine-related transcripts a history of alcohol drinking and abstinence causally alters the likelihood that expression × function correlations emerge.

More research will certainly be needed to elucidate the mechanisms underlying the observed changes in synchrony between transcript expression and function. Experience-dependent shifts in a plethora of biological processes such as alterations in rate of protein translation, post-translational modifications, and trafficking could potentially explain these results. For example, a higher level of DNA transcription may not be reflected in RNA read counts if the RNA is being translated to protein at a faster rate, resulting in a decorrelation between RNA counts and protein expression. This could occur through a range of mechanisms including rate of 5′ capping, mRNA stability, or localization. Even in cases where a given gene is primarily regulated at the level of transcription and RNA levels are correlated with protein expression, if post-transcriptional alterations, such as RNA editing or alternative splicing, dictates protein activity, transcript expression × function relationships may not be present. Likewise, any number of post-translational modifications could alter functional outputs independent of transcript expression, through differential trafficking, affinity states, and rates of protein degradation. These examples only constitute a small number of possible explanations.

Surprisingly, not only did we observe that experience-dependent plasticity in the presence or directionality of transcript expression × function relationships appear to be the rule rather than the exception, but the control condition was most associated with a lack of a correlation between transcript expression and functional output. It is intriguing to speculate that in an alcohol-naïve system, dopamine release, reuptake and regulation may be primarily through fast, local mechanisms at the presynaptic terminal, such that a full range of activity can be achieved with a minority of the available receptors and transporters (i.e. operating in a 'spare receptor reserve' mode); on the other hand, the emergence of expression × function relationships after chronic alcohol and protracted abstinence, despite no change in transcript levels, may indicate that the system is operating under conditions where the range of cellular activity is dictated by the availability of protein present at synaptic sites at a given time (i.e. in a 'spare receptor depleted' mode). If this is the case, it would likely produce a restricted range of function (e.g. Fig. 3) and a much less plastic system, even though there may be no observable change in transcript expression or function in a cross-sectional measurement. Given the presence of wide-spread alterations in transcript expression

× function relationships observed across protein classes in protracted abstinence, it is important to consider whether these changes in synchrony may contribute to inflexibility of dopaminergic signaling to precipitate relapse-related behaviors.

Finally, this work was exclusively conducted in male subjects, and it will thus be critical to perform parallel investigations in female subjects to determine whether this plasticity and modulation of transcription × function relationship is consistent across sexes. Nonetheless, here we show that alcohol intake induces long-lasting functional changes that are present after protracted abstinence and are characterized by a reorganization of the relationship between gene expression and functional measures of dopamine terminal activity. In an animal and experimental model with great translational relevance, this work offers insights into the biological changes during abstinence, a crucial timepoint for therapeutic intervention for AUD, and highlights the complexity of a biological relationship often taken for granted.

## Methods

### Subjects
Subjects were 28 male rhesus macaques (*Macaca mulatta*) between the ages of 7 and 8.3 years across two cohorts ("Rhesus 10" [alias: Cohort 1] and "Rhesus 14" [alias: Cohort 2], cohort details at www.matrr.com). Weights of the subjects ranged from 7 to 12.6 kg. Animals were individually housed in quadrant cages (0.8 × 0.8 × 0.9 m) with constant temperature (20–22 °C) and humidity (65%) and an 11 h light cycle (lights on at 8:00 AM). Animals had visual, auditory, and olfactory contact with other conspecifics, in addition to 2 h of pair housing each day. Body weights were measured weekly. All procedures were conducted in accordance with the Guide for the Care and Use of Laboratory Animals and approved by the Oregon National Primate Research Center Institutional Animal Care and Use Committee.

### Drinking procedure
Monkeys (17 alcohol drinkers and 11 calorically yoked or housing controls) were trained to obtain fluids and their meals from an operant panel that replaced one of the walls of their home cage, as described previously[31,40]. Briefly, the panels had two spouts, one to each side of a 15-inch video display screen. Near each spout, the display showed a set of three stimulus lights (white, red, and green) that indicated an active session or food or fluid availability, respectively. A centrally located recessed dowel activated the fluid spouts, and an infrared finger poke activated the pellet dispenser (env-203-1000; Med Associates). Each spout was connected via Nalgene tubing to a 1 L fluid reservoir set on a digital scale (Adventurer Pro Balances AV4101C; Ohaus). Dowel pulls, finger pokes, and fluid consumption were recorded every 500 ms via a computerized system (Dell Optiplex) using custom hardware and programming using a National Instruments interface and Labview software. Schedule-induced polydipsia, as described previously[31,40], was used to induce alcohol self-administration in daily 16 h sessions. Briefly, a 1 g banana food pellet (Research Diets) was dispensed every 300 s (fixed time, 300 s) until a water volume equivalent to 1.5 g/kg of 4% (w/v) ethanol was consumed. Following at least 30 days of water induction, 4% ethanol replaced water. In 30-day increments, each animal consumed increasing doses of 4% ethanol: 0.5 g/kg/d, 1.0 g/kg/d, then 1.5 g/kg/d. Following consumption of the programmed volume, water was immediately available, and any remaining pellets were available on a fixed-ratio 1 (FR-1) schedule after a 2 h time-out. Following completion of ethanol induction, daily 22 h open access sessions were performed, during which water and ethanol were concurrently available. Food pellets were available on a FR-1 schedule in at least three daily meals in 2 h intervals starting at the session onset. Data were downloaded, husbandry tasks were performed, food and fluids were replenished, and fresh fruit was provided each day by technicians during the 2 h break.

After drinking was initiated, subjects had 12 months of open access to alcohol for 22 hours a day. 6 months into this period, an endocrine profile was collected. After 12 months of alcohol access, subjects underwent a one-month abstinence period. Following this abstinence period, subjects returned to open access alcohol exposure for three months, then underwent another one-month withdrawal period, returned to three months of open access alcohol exposure, and finally underwent one month of withdrawal at the end of which subjects were necropsied. During open access, blood was collected weekly for blood ethanol concentration measurements. Behavioral data collection and analysis was blind to voltammetry and RNA-seq results.

### Tissue preparation
Tissue preparation was described previously[37,41]. Briefly, monkeys were anesthetized with ketamine (10 mg/kg), maintained on isoflurane, and perfused with ice-cold oxygenated monkey perfusion solution [containing (in mM) 124 NaCl, 23 NaHCO3, 3 NaH2PO4, 5 KCl, 2 MgSO4, 10 d-glucose, 2 CaCl2]. Brains were quickly removed and 4-6 mm sections were made along the coronal plane using a brain matrix (Electron Microscopy Sciences), with the brain knife position guided by each individual's MRI. An isolated tissue block containing only the striatum (caudate, putamen, and nucleus accumbens) was placed in ice-cold oxygenated monkey perfusion solution and transported on ice for slicing.

### In vitro voltammetry
Fast-scan cyclic voltammetry was then used to characterize presynaptic dopamine release and uptake as well as the ability of kappa opioid receptors to decrease dopamine release, or kappa opioid receptor sensitivity, in the NAc core. Voltammetric detection of dopamine in brain slices has been used by the authors and others to examine receptor regulation of dopamine release and uptake kinetics[29,85,86]. A ceramic blade attached to a vibrating tissue slicer was used to prepare 250-μm-thick coronal brain sections containing the NAc core. The tissue was immersed in oxygenated artificial CSF (aCSF) containing the following (in mM): 126 NaCl, 2.5 KCl, 1.2 NaH2PO4, 2.4 CaCl2, 1.2 MgCl2, 25 NaHCO3, 11 glucose, and 0.4 l-ascorbic acid, pH adjusted to 7.4. Once sliced, the tissue was transferred to testing chambers containing bath aCSF (32 °C), which flowed at 2 ml/min. A carbon fiber microelectrode (50–150 μm length, 7 μm diameter) and bipolar stimulating electrode were placed in close proximity on the tissue. Extracellular dopamine was recorded by applying a triangular waveform (−0.4 to +1.2 to −0.4 V vs Ag/AgCl, 400 V/s) to the recording electrode and scanning every 100 ms. This waveform allows for the assessment of oxidation and reduction peaks for dopamine, and has been used extensively to detect dopamine in brain[44]. Dopamine release was evoked by 1 pulse stimulations (350 μA, 4 ms, monophasic) applied to the tissue every 5 min, based on the shared $EA_{50}$ between drinkers and controls. When a stable baseline was established (three collections within 10% variability) and predrug measures were taken, the selective kappa opioid receptor agonist U50,488 (0.3 μm and 1 μm, cumulatively) was bath applied to the slice, and stimulations continued until stability was reached at each concentration.

### Voltammetry analysis
For all acquisition and analysis of FSCV data, Demon voltammetry and analysis software was used[87]. Recording electrodes were calibrated by recording responses (in electrical current; in nanoamperes) to a known concentration of dopamine (3 μm) using a flow-injection system. This was used to convert electrical current to dopamine concentration. Baseline recordings (i.e., after stabilization criteria were met but prior to drug application) FCSV data were modeled using Michaelis–Menten kinetics ($K_M$ set to 160 nM), which allows for the determination of evoked dopamine release and the maximal rate of dopamine uptake

($V_{max}$). Investigators were blind to group assignment during data collection and analysis of voltammetry data.

## Tissue preparation and RNA-seq
RNA was extracted from frozen tissue biopsies using the AllPrep DNA/RNA/miRNA Universal Kit (Qiagen Sciences, Germantown, MD, USA) following manufacture's protocol by the ONPRC Primate Genetics Core. RNA integrity was confirmed with a 2100 Bioanalyzer (Agilent Technologies, Palo Alto). Samples were sequenced on Illumina NovaSeq 6000. The Vanderbilt Creative Data Solutions Shared Resource (RRID:SCR_022366) assisted with bulk RNAseq preprocessing and analysis. Paired-end RNA sequencing reads (150 bp long) were trimmed and filtered for quality using Trimgalore v0.6.7 (Krueger et al. 2021). Trimmed reads were quantified using Salmon[88] v1.9.0 with the Mmul10 *Macaca mulatta* genome. Sample read counts were normalized using DESeq2 v1.36.0[89]. Features counted fewer than 5 times across at least 3 samples were filtered. For pairwise comparisons aligned read counts were analyzed for differential gene expression using the Biojupies analysis package with default settings[90].

## Weighted gene correlation network analysis (WGCNA)
We performed weighted correlation with individual sample weights determined with the 'signed hybrid' network (where negatively correlated genes are assumed not connected)[91]. The soft thresholding power was determined using scale-free topology of each sample as a fit index. From the determined scale of independence and mean connectivity calculated, we used a soft thresholding power of eight to perform WGCNA. To identify unique modules, a one-step network was constructed using blockwise modules constructed with unsigned topological overlap matrices. To identify distinct modules, we utilized the Dynamic Tree Cut method. Of note, only module eigengenes that reached a threshold of 0.99 or above were included for subsequent network analysis. The edge and node data for all modules were exported to the external R package cytoHubba[92] to determine significant hub objects via the topological analysis method, Maximal Clique Centrality (MCC). The top 25 eigengenes with the (MCC) scores for a specific module were visualized using Cytoscape software to highlight important hubs and for ease of visualization[93].

To determine the relationship between modules, the Pearson correlation coefficients between module eigengenes was calculated. Similarly, the relationship between individual eigengenes and with treatment ('controls' versus 'drinkers') were calculated using a Pearson correlation. Heatmaps and pathway analyses for WGCNA figures included all genes assigned to a specific module.

## Drugs
CMPD101 was obtained from Hellobio. NEM, $BaCl_2$, U50,488, and NorBNI were received from Sigma Aldrich, and nalfurafine was received from Fisher. Each drug was made fresh as a stock solution at 1 mM for NorBNI, or 10 mM for U50,488, CMPD101, NEM, and $BaCl_2$. Stock solutions were then added to the aCSF reservoir to reach the final concentration.

## Statistics
Statistical analyses were performed using GraphPad Prism (V10). For all pairwise comparisons between two conditions or groups, we utilized paired or unpaired *t*-tests, respectively. Comparisons across three or more variables were made using one-way ANOVAs or two-way ANOVAs (followed by Šídák's multiple comparisons when planned comparisons were made or interactions were detected). For correlation analyses, Pearson's correlation coefficient was used for continuous variables and Spearman's correlation coefficient was used for ordinal variables. In both cases, correlations were performed within-group only, as required to meet assumptions of bivariate normality in correlative analyses[94–97]. Primary fast-scan cyclic voltammetry measures were collected from Cohort 1, thus all voltammetry x gene expression correlations only include that subset of subjects. Cohort 2 was used for follow-up fast-scan cyclic voltammetry experiments and therefore did not have the correlated measured recorded. Bulk gene expression and drinking behavior x gene expression analyses included both cohorts. All tests were two-tailed and *p* values <0.05 were considered to be statistically significant.

## Reporting summary
Further information on research design is available in the Nature Portfolio Reporting Summary linked to this article.

## Data availability
The sequencing data generated in this study have been deposited in the GEO database under accession code GSE244557. Source data are provided with this paper.

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

## Acknowledgements

This work was supported by NIH grants R00 DA04510 (C.A.S), R01 AA030115 (C.A.S.), U01 AA029971 (C.A.S.), U01 AA013510 (K.A.G.), R24 AA019431 (K.A.G.), U24 AA013641 (K.A.G.), P51 OD0119092 (K.A.G.), U01 AA014091 (S.R.J., K.M.H.), P50 AA026117 (S.R.J), T32AA007565 (K.M.H, S.C.F), and T32DA041349 (S.R.J.) as well as through the Alkermes Pathways Research Award (C.A.S.), the Brain Research Foundation (C.A.S.), and the Whitehall Foundation (C.A.S). Z.Z.F was supported by an NIH fellowship (DA056196).

## Author contributions

Conceptualization: Z.Z.F., S.C.F., K.A.G., S.R.J., C.A.S.; Data curation: Z.Z.F., K.A.G., C.A.S.; Formal analysis: Z.Z.F., K.M.H., J.P.S., D.D.K., K.A.G., C.A.S.; Funding acquisition: K.M.H., V.C.C.C., K.A.G., S.R.J., C.A.S.; Investigation: K.M.H., S.C.F., M.I.M., A.J.L., V.C.C.C., K.A.G., C.A.S.; Methodology: K.A.G., S.R.J., C.A.S.; Project administration: V.C.C.C., K.A.G., S.R.J., C.A.S.; Resources: V.C.C.C., K.A.G., S.R.J., C.A.S.; Supervision: K.A.G., S.R.J., C.A.S.; Validation: Z.Z.F., J.P.S., D.D.K., K.A.G., C.A.S.; Visualization: Z.Z.F., J.P.S., D.D.K., C.A.S.; Writing—original draft: Z.Z.F., J.P.S., D.D.K., C.A.S.; Writing—review & editing: Z.Z.F., K.M.H., J.P.S., S.C.F., M.I.M., A.J.L., V.C.C.C., D.D.K., K.A.G., S.R.J., C.A.S.

## Competing interests

The authors declare no competing interests.
