## [Transparent Peer Review file · Nature Communications]

Synchrony between midbrain gene transcription and dopamine terminal regulation is modulated by chronic alcohol drinking

Corresponding Author: Dr Cody Siciliano

Version 0:

Reviewer comments:

Reviewer #1

(Remarks to the Author)

The study by Farahbakhsh and colleagues investigates the persistence of alcohol-induced alterations in dopamine transmission after a month-long abstinence from chronic repeated alcohol drinking in non-human primates. More specifically, this study focuses on dopamine reuptake and the expression of the dopamine transporter, as well as the kappa opioid receptor regulation of dopamine release, which have been previously shown to be altered in rodents shortly after alcohol drinking. This study was performed on a large cohort of rhesus macaques that consumed alcohol for over a year with an interrupted 3-month period of forced abstinence. Animals were studied after 1-month of forced abstinence from the last exposure and compared to water and caloric-matched controls.

The results show overall smaller dopamine (DA) signals in alcohol drinkers (data in supplemental Figure) and faster reuptake and higher sensitivity to kappa agonists in drinkers compared to controls, even after this month-long abstinence. These results support the idea that these alterations previously described in rodents and seen also in macaques are persistent. The study also evaluates the expression level of genes related to dopamine reuptake and kappa opioid receptors in the midbrain of these animals and correlates it with the functional analysis. Interesting observations are made about positive and negative correlations between the functional readouts in the striatum and the gene expression in the midbrain where dopamine neuron cell bodies are located.

The study is carefully done and controlled. The results are interesting and carry further relevance by providing some extension from results found previously in rodents.

Main comments:

1. FSCV data in Suppl. Fig 3 shows dopamine signals are smaller in drinkers than controls. Why isn't this in the main figure? Also, please explain why the n is 3 and 7 here and larger in the data from figure 3?
2. Are the changes in max DA and update correlated to each other?
3. As stated by the authors, these genes tested, which expression are correlated with DA release and reuptake, encode for proteins that are expressed at the axons of DA neurons (OPRK1, DRD2, DAT, VMAT). Would it be possible that DA axon innervation is denser in drinkers than controls? Can this be tested or rule out with immunocytochemistry or an anatomical analysis?
4. Fig. 2 correlations. Can correlation be tested with both groups combined? I don't see the rationale for separating the groups if there is in fact a correlation with all groups together. At least, this should be tested. My concern is that the small sample size in the control group is preventing from identifying significant correlations in some cases. Besides, the range of change in the control is smaller. Having both groups together could strengthen the correlation and further make the point that changes in genes expression in drinkers are associated with the functional changes in DA transmission.
5. The rank analysis makes a quite compelling point for the lack of correlation in the control. Still, it would be important to see the rank analysis done with all the data together, if possible.

6. Fig 4. Correlations. could the authors probe the receptor/ligand ratio for the individual animals and how these relates to the inhibition by the exogenously applied ligand?
 7. Fig. 4. pDYN. Is the expression also assessed in the VTA area? Is the expression from D1-MSN terminals? Clearly, there is evidence of KOR ligand release during the train stimulation in the NAc of drinkers. More clarification on this matter would be appreciated.
- Other minor points:
8. Rank order plots in Fig. 2 and 4. Could you please add the parameter measured in x-axis to the axis label. E.g.: "dopamine release" or "kappaR mediated inhibition", etc...
 9. Fig. 2. Are panel D-G from NAc or VTA? Please label more clearly and add units to the y-axis.
 10. Fig. 1. Are the colors identifying unique animals? Could authors connect the dots that correspond to the same animal? Panel E -y axis units missing
 11. Figure 3. panels D-K. what is the unit and the measurement of the y-axis?
 12. Fig. 5 should data from panels C and F be presented together and stats compared that way?
 13. Fig. 5 C and F. y-axis label is unclear. Which is the frequency being plotted? Is this an average of all frequencies?
 14. Fig. 5. what about changes in gene expression in the NAc?

Reviewer #2

(Remarks to the Author)

The manuscript explores the relationship between gene expression changes in the ventral tegmental area and dopamine dynamics in the nucleus accumbens in male rhesus macaques with an extensive alcohol drinking history interspersed with three forced abstinence periods. The measurements were done after the final abstinence period and thus the authors are examining changes that persist into abstinence, which is highly relevant to understanding persistent brain and behavior changes in alcohol use disorder (AUD). The authors hypothesized that this alcohol drinking history would induce a dysregulation in the expression of dopamine neuron-specific genes in the VTA. The authors found that in the full cohort (n=17) this alcohol drinking and abstinence history did not induce changes in gene expression (using full transcriptomic RNA-sequencing) in the VTA relative to the controls (n=11). In parallel to the gene expression analyses, 8 macaque drinkers and 8 controls were used to assess dopamine dynamics in the nucleus accumbens dopamine terminals using voltametric detection. However, it is not clear how these subjects were selected from the larger cohort. Here the authors found increased dopamine reuptake in the subjects with the alcohol drinking/abstinence history, but no change in dopamine release at the 350 uA stimulation (though decreased dopamine release was evoked by high intensity stimulations). Next, the authors examined the relation between gene expression in the VTA and accumbal dopamine dynamics using correlation analyses. The analysis extended to genes that encode for regulators of dopamine release such as the kappa opioid receptor (OPRK1) and others (DRD2, SLC6A3, SLC18A2). In the drinkers it was found that DRD2 and OPRK1 positively correlated with dopamine release in the nucleus accumbens. This correlation was not observed in the controls. This led to a focus on kappa opioid receptor which may be an important target for therapeutic development for the treatment of alcohol use disorder. Overall, the focus on this extensive alcohol drinking and abstinence is important for the field. There are numerous strengths of this work including the replication of the literature showing a hypodopaminergic state following an extensive alcohol history, associations between gene expression and dopamine release, inclusion of pharmacological manipulation of kappa opioid receptor (KOR) on dopamine function in the nucleus accumbens to delve into the functional changes, and using a highly translational model. My overall concern is that the major findings of this study are corollary. Alcohol drinking history did affect accumbal dopamine dynamics which is consistent with the literature, so that is good that it was confirmed, but this is not new information, and gene expression in the VTA was unchanged by drinking history. Therefore, it feels like a stretch to then be examining correlations between gene expression changes and accumbal dopamine. The idea that synchrony between functional release and upstream transcription is disrupted is not entirely compelling and feels like an overstatement. How are these changes functionally related? Other concerns are described below.

1. The description of the results from Figure 1 set up the reader for the expectation that there was going to be a focus on individual differences. To this end, there is no description of why the gene expression analyses were conducted in all the subjects, but only 8 from each group were selected for the recordings. A description of the justification for which subjects were selected is necessary as is defining what their alcohol drinking levels were.
2. There were no changes in gene expression following a history of alcohol drinking/abstinence in the full cohort. However, were there any changes in the smaller subgroup that was used in the majority of the paper? Relatedly, it is conceptually confusing that drinking history doesn't affect gene expression changes but yet, the focus of the paper is on correlations between gene expression and dopamine dynamics.
3. The functional studies of this manuscript focused on the effect of exogenous activation of the kappa opioid receptor system on dopamine release dynamics in the nucleus accumbens. Again, results confirmed previous work. It will be important to highlight what aspect of these studies and the results that are novel.

4. This study was conducted in male subjects. While I understand the limitations, given the review recommendations of the journal to consider sex in the area of study, it will be important to address this in the manuscript.

5. I would recommend that the authors clarify that the gene expression data is from the VTA and the dopamine measures are from the accumbens – clarifying this especially in the figures would be helpful. This also applies to statements of hypotheses and conclusions. It is not that alcohol-induced changes in gene expression were not observed, but rather they were not observed in the VTA.

Reviewer #3

(Remarks to the Author)

I co-reviewed this manuscript with one of the reviewers who provided the listed reports. This is part of the Nature Communications initiative to facilitate training in peer review and to provide appropriate recognition for Early Career Researchers who co-review manuscripts

Version 1:

Reviewer comments:

Reviewer #1

(Remarks to the Author)

The revisions include clarifications and additional analyses, which are mainly included in the rebuttal letter, not the manuscript. Additionally, the authors have only partially addressed my comments. Here are a few examples:

Comment 2: The authors addressed the second part of the question but did not explain why the voltammetry input-output curve is not shown in the main figure. To be more direct, I request that the authors display the voltammetry input-output curve in Figure 3.

Comment 3: My question was: "Are the changes in max DA and reuptake correlated to each other?"

The authors responded with a correlation using DA amplitudes that are not maximal, and they used EA50 instead.

- a) This plot does not answer my comment.
- b) The plot shows a positive correlation between reuptake speed and dopamine release, which contradicts the findings of the study. The results show faster reuptake and lower max release.
- c) I assume the EA50 is the same intensity for both groups. Could the authors please confirm this? Otherwise, these are not direct comparisons.

There are also other contradictions throughout the manuscript that are hard to reconcile. For example, the results section states: "However, the maximal rate of dopamine reuptake (V_{max}) was increased in subjects with a history of chronic alcohol self-administration (Figure 3C), suggesting an upregulation of dopamine transporter activity and faster removal of dopamine from the extracellular space, thereby decreasing tonic extracellular dopamine levels." However, the study does not measure tonic dopamine levels, but rather evoked dopamine release. This discrepancy needs to be clarified.

Additionally, in the rebuttal letter, the authors use an opposing argument to address comment 3: "Essentially, when uptake rate is high, there is less dopamine outside and thus more dopamine inside the terminal, which leads to high release as well."

This contradiction between the results section and the rebuttal response requires resolution.

Comment 4: The response could be improved. The authors argue that while the suggestion is valid, they have already used the available brain tissue. However, they have access to either fixed or fresh tissue from alcohol drinkers, which could be used for immunostaining or western blot analysis. I recommend considering this option.

Comment 7-8: I appreciate the additional analysis for correlation of DYN and kappa receptor expression, as well as the clarification on the tissue collection. A remaining question is: How do authors interpret the negative correlation between kappa receptor expression in VTA and inhibition of DA release in the Nac, the fact that more inhibition is seen when the receptor expression is lower? This is counterintuitive.

Reviewer #2

(Remarks to the Author)

I really appreciate the care with which the authors responded to my comments and the thorough explanations. The clarifications/additions in the text are also very helpful. My concerns were all addressed and I have no additional

comments/concerns.

Reviewer #3

(Remarks to the Author)

Version 2:

Reviewer comments:

Reviewer #1

(Remarks to the Author)

The revised manuscript includes further revisions that improve the manuscript, now showing and interpreting the findings more accurately. I appreciate the additional responses and explanations.

We would like to thank the reviewers and editors for their time and constructive comments on our manuscript. At the request of the reviewers, we have added multiple new analyses including cohort comparisons of drinking data, causal statistical tests to compliment the correlational analyses in of kappa receptor sensitivity, and receptor/ligand ratio to receptor sensitivity correlations. Further, we have restructured and expanded our discussion of the expression \times function correlations, and clarified the experimental and statistical design both in diagrams and text throughout. We are grateful for their insightful suggestions and we believe these changes have considerably strengthened the impact, rigor, and clarity of the manuscript.

Responses to Individual Reviewer Comments:

Please find point-by-point responses to each reviewer's comments, colored coded by reviewer:

Responses to Reviewer #1

Responses to Reviewer #2

Responses to Reviewer #3

A clean copy of the revised manuscript is provided, as well as a version with changes denoted in blue.

Reviewer #1 (Remarks to the Author):

Comment 1: The study by Farahbakhsh and colleagues investigates the persistence of alcohol-induced alterations in dopamine transmission after a month-long abstinence from chronic repeated alcohol drinking in non-human primates. More specifically, this study focuses on dopamine reuptake and the expression of the dopamine transporter, as well as the kappa opioid receptor regulation of dopamine release, which have been previously shown to be altered in rodents shortly after alcohol drinking. This study was performed on a large cohort of rhesus macaques that consumed alcohol for over a year with an interrupted 3-month period of forced abstinence. Animals were studied after 1-month of forced abstinence from the last exposure and compared to water and caloric-matched controls.

The results show overall smaller dopamine (DA) signals in alcohol drinkers (data in supplemental Figure) and faster reuptake and higher sensitivity to kappa agonists in drinkers compared to controls, even after this month-long abstinence. These results support the idea that these alterations previously described in rodents and seen also in macaques are persistent. The study also evaluates the expression level of genes related to dopamine reuptake and kappa opioid receptors in the midbrain of these animals and correlates it with the functional analysis. Interesting observations are made about positive and negative correlations between the functional readouts in the striatum and the gene expression in the midbrain where dopamine neuron cell bodies are located.

The study is carefully done and controlled. The results are interesting and carry further relevance by providing some extension from results found previously in rodents.

Response 1: We thank the reviewer for their time and expertise and appreciate the recognition of the importance of the work. See **Reviewer 2 Response 1** for an explanation of why we believe this work to go far beyond replicating rodent findings.

Main comments:

Comment 2: FSCV data in Suppl. Fig 3 shows dopamine signals are smaller in drinkers than controls. Why isn't this in the main figure? Also, please explain why the n is 3 and 7 here and larger in the data from figure 3?

Response 2: Subjects/tissue are very limited, thus we carefully plan the number of slices dedicated to each experiment. While we were able to get basic measures of release and reuptake from the baseline measurements prior to any drug application, the input/output experiments take considerably more time (we only have ~12-16 hours of viable recording time per animal). Thus, experiments in **Supplemental Figure 3** were matched to prior sample sizes which were sufficient to detect group differences in release and reuptake present at zero withdrawal time (Siciliano et al., 2015, 2016, 2018), and these data were used to determine the EA_{50} of electrically evoked dopamine release, which did not differ between groups and was used for the remainder of the manuscript. This rationale has now been clarified in the methods and results section and a comparison of EA_{50} values between drinkers and controls has been added to **Supplemental Figure 3**.

Comment 3: Are the changes in max DA and uptake correlated to each other?

Response 3: In general, yes, release and reuptake often covary (Wightman and Zimmerman 1990; Ferris et al. 2013). Work from many groups has shown that DAT is a primary determinant of tonic extracellular dopamine levels, and that repackaging of dopamine via reuptake is important for vesicular content, and the reliance of dopamine release on repackaging via the DAT causes release and uptake to be closely related in many situations (Ferris et al., 2014; Siciliano et al., 2014). Essentially, when uptake rate is high,

Figure R1. (A) Correlation between dopamine release and maximal rate of dopamine uptake (V_{max}) for drinkers and controls with the best-fit linear regression shown and r and p values shown as inset.

there is less dopamine outside and thus more dopamine inside the terminal, which leads to high release as well. Release is not completely reliant on reuptake, so there are certainly situations in which release and uptake are not correlated, or even changed in opposite directions, but more often they are related, as the reviewer has suggested. In the current dataset, we also see this relationship (**Figure R1**).

Comment 4: As stated by the authors, these genes tested, which expression are correlated with DA release and reuptake, encode for proteins that are expressed at the axons of DA neurons (OPRK1, DRD2, DAT, VMAT). Would it be possible that DA axon innervation is denser in drinkers than controls? Can this be tested or rule out with immunocytochemistry or an anatomical analysis?

Response 4: The reviewer brings up an interesting question that could potentially explain some differences between drinkers and controls. It is certainly possible that dopamine axon innervation in the nucleus accumbens is altered with a history of ethanol intake. Though we are not aware of a direct test of this possibility, several pieces of evidence point towards a potential *decrease* in dopaminergic innervation of the striatum following chronic alcohol exposure. For example, in rats, chronic alcohol decreases tissue content of dopamine and its primary metabolites (Rothblat et al., 2001) and there is some evidence that chronic alcohol exposure can increase diffusion of neurotransmitters, including dopamine, which suggests decreased overall density of structures in the extracellular space (De Santis et al., 2020). Further, decreased dopamine axon density in the nucleus accumbens prior to first drink is associated with heightened alcohol preference (Zhou et al., 1995). Importantly, while we cannot count out differences in innervation, denser innervation would likely be associated with increased dopamine release magnitude; here, we see no change in response to the half-maximal stimulation intensity and a decrease in the overall range of dopamine release across the input/output curve. While we agree that assessing density of dopamine innervation in the nucleus accumbens between drinkers and controls would certainly be interesting, it is unfortunately not possible in these subjects because the slices from the nucleus accumbens were used for fast-scan cyclic voltammetry recordings and are thus not available for protein analysis.

Comment 5.1: Fig. 2 correlations. Can correlation be tested with both groups combined? I don't see the rationale for separating the groups if there is in fact a correlation with all groups together. At least, this should be tested. **5.2:** My concern is that the small sample size in the control group is preventing from identifying significant correlations in some cases. Besides, the range of change in the control is smaller. Having both groups together could strengthen the correlation and further make the point that changes in genes expression in drinkers are associate with the functional changes in DA transmission.

Response 5.1: Combined correlations on data pooled across controls and drinkers would violate assumptions of bivariate normality in correlative analyses (see Simpson's Paradox/Yule-Simpson effect). Violating these assumptions would be especially problematic in this case because we show that significant between-group differences are detected in one of the measures that are being correlated (reuptake) (Janse et al., 2021; Pearson, 1896; Simpson, 1951; Yule, 1903). We have provided the graphical analysis and regressions for the reviewer here (**Figure R2 and R3**) and have clarified the necessity of reporting within-group correlations only in the methods section of the manuscript.

Response 5.2: While we agree that there is of course some chance of type II error, it is important to note that we are not relying on the null correlations in the control subjects for our conclusions *per se*; rather, the conclusions were drawn from the statistical contrast in the mean slopes across many regressions in the controls and drinkers, which were significant different for the *a priori* selected genes (**Figure 3M, Figure 4M**), but did not differ in the matched control gene set (**Supplemental Figure 4I**). Contrasting the mean slope values is valid because the sample sizes are matched between the controls and drinkers (8 gene analyses derived from 8 subjects per group), and this method allows for inference of causality regarding alcohol-induced changes in the relationship between gene expression and function. Furthermore, and speaking more directly to the reviewers' concerns, when we normalized values to allow for ordinal correlations across 64 x-y pairs per group, control subjects displayed even lower r values, approaching zero ($r=0.07$), while drinkers retained a robust positive correlation ($r=0.75$) (**Figure 1M-N**).

Figure R3. Scatter plots and across-group regressions of accumbal dopamine release magnitude and VTA expression of heterosynaptic genes. Expression of genes encoding for receptors and transporters in the VTA which are enriched in non-dopaminergic cells was compared to terminal dopamine release and reuptake in the NAc of the same subjects. The best-fit linear regressions are plotted. **(A-D)** When groups were pooled, none of the genes selected correlated with dopamine release, R^2 values are shown in inset and were 0.06 or lower for **(A)** the dopamine receptor 1 (*DRD1*), **(B)** the dopamine receptor 5 (*DRD5*), **(C)** the glutamate transporter 2 (*VGLUT2*), **(D)** and the vesicular GABA transporter (*VGAT*). **(E-H)** The relationship between each subject's average V_{max} and upstream gene expression of homosynaptic regulators of dopamine terminals was assessed. The best-fit linear regression and R^2 are shown. When controls and drinkers are pooled, there is no correlation with upstream gene expression for **(E)** *DRD1*, **(F)** *DRD5*, **(G)** *VGLUT2*, **(H)** and *VGAT*.

Comment 6: The rank analysis makes a quite compelling point for the lack of correlation in the control. Still, it would be important to see the rank analysis done with all the data together, if possible.

Response 6: See **Response 5.1** for an explanation of why we think it is not statistically appropriate to include across-group correlations in the manuscript. However, we have provided the graphical representations of the analysis here (**Figure R4**), and would be happy to reconsider if an alternative line of reasoning is presented by the reviewers or editors.

Comment 7: Fig 4. Correlations. could the authors probe the receptor/ligand ratio for the individual animals and how these relates to the inhibition by the exogenously applied ligand?

Response 7: We agree that this is a worthwhile addition as the receptor/ligand ratio has previously shown unique alcohol experience-dependent relationships (Bazov et al., 2018), and receptor sensitivity is often associated with endogenous ligand levels. The relationships and linear regressions are shown below (**Figure R5**) and this figure has thus been added as **Supplemental Figure 9** and is now referred to in the text. There was no relationship between *PDYN/OPRK1* expression and kappa-mediated inhibition in either drinkers or controls. This is likely because though the VTA is a source of dynorphin, the primary source in the accumbens is thought to be local D1-MSN cell bodies (as the reviewer mentioned) and that source would not be captured in our sequencing data. Our data also suggest, albeit indirectly, that the alcohol-induced increases in kappa opioid receptor sensitivity may be via alterations in β -arrestin activity. This provides a potential explanation as to why there is an apparent decoupling of the typical relationship between extracellular ligand and receptor signaling, as shown in prior work in tissue homogenates following ethanol vapor exposure (Kissler et al., 2014) and here with increased receptor sensitivity and dynorphin release probability.

Comment 8: Fig. 4. pDYN. Is the expression also assessed in the VTA area? Is the expression from D1-MSN terminals? Clearly, there is evidence of KOR ligand release during the train stimulation in the NAc of drinkers. More clarification on this matter would be appreciated.

Response 8: We apologize for the lack of clarity. The expression levels assessed here are indeed from the VTA. All RNA sequencing experiments were conducted on bulk samples from the VTA and all voltammetry experiments were performed in the accumbens – we have now clarified this throughout the text, and modified the experimental diagram in Figure 1 to ensure that this is clearly conveyed.

Regarding PDYN, it is possible that locally translated mRNA in D1 MSN terminals in the VTA could be contributing to some of the observed signal, but it is unlikely to be a major contributing factor given that dynorphin is expressed in several cell types in the VTA (Phillips et al., 2022). Several of these cell types project from the VTA to accumbens (Swanson, 1982), which may be contributing to the observed relationship between VTA PDYN expression and functional measures of dynorphin release probability in the NAc (**Figure 5H**). Given the upregulation of putative dynorphin release probability directly in the NAc, future work will also aim to investigate potential transcriptional regulation of PDYN in post-synaptic medium spiny neurons.

Other minor points:

Comment 10: Rank order plots in Fig. 2 and 4. Could you please add the parameter measured in x-axis to the axis label. E.g.: “dopamine release” or “kappaR mediated inhibition”, etc...

Response 10: We apologize for the lack of clarity. Y-axes now state “Within-Group Rank Expression (relevant gene names)” and the x-axes now state “Within-Group Rank Function (relevant functional measures)” in Figure 3, Figure 4, and Supplemental Figure 4.

Comment 11: Fig. 2. Are panel D-G from NAc or VTA? Please label more clearly and add units to the y-axis.

Response 11: All the sequencing data, including the whole of **Figure 2B** is from the VTA. We agree with the reviewers that the manuscript lacked clarity on that detail and have reiterated it for clarity across a number of figures, legends, and within the text. Units have been added to the y-axis and the figure legend, the colors of the heatmap indicate Pearson's r values.

Comment 12: Fig. 1. Are the colors identifying unique animals? Could authors connect the dots that correspond to the same animal? Panel E -y axis units missing

Response 12: Yes, the colors are identifying unique animals and the same color refers to the same subject across the figure – this has now been clarified in the figure legend. The dots corresponding to the same animal have been connected in **Figure 1C**. The unit for blood alcohol content has been added to the axis title in **Figure 1E**.

Comment 13: Figure 3. panels D-K. what is the unit and the measurement of the y-axis?

Response 13: We apologize for the lack of clarity. The unit of measurement is variance stabilized counts of each of the genes of interest (Love et al., 2014). This label has been added to the y-axis where relevant.

Comment 14: Fig. 5 should data from panels C and F be presented together and stats compared that way?

Response 14: Originally, we were not aiming to compare drinkers to controls but rather look within-group to determine whether NorBNI increased dopamine release, thus they were analyzed separately. But we agree with the reviewer that the way the results are discussed, it would be useful to include an omnibus test between groups as well. We have changed the statistical test to a mixed-model two-way ANOVA with Sidak's multiple comparisons test between NorBNI and baseline measures and this change has been outlined in the figure legend (**Figure 5**). Of note, the findings do not change.

Comment 15: Fig. 5 C and F. y-axis label is unclear. Which is the frequency being plotted? Is this an average of all frequencies?

Response 15: The y-axis is the area under the curve (AUC) of normalized dopamine release across the whole frequency curve (from 5 to 100 Hz) which corresponds to the areas of the shaded regions in B and E. The y-axis label has been adjusted to say "AUC (dopamine release x frequency)" to improve clarity of this measure.

Comment 16: Fig. 5. what about changes in gene expression in the NAc?

Response 16: This study only assessed gene expression in the VTA because we were particularly interested in the potential mechanisms by which long-term changes in the function of the presynaptic dopamine terminals themselves might persist throughout abstinence. While the NAc gene expression changes would be interesting and potentially important regarding the dynorphin release probability findings, they are unlikely to be driving the presynaptic changes. Therefore, we conducted the sequencing in the compartment which the cell bodies of these terminals are housed (i.e., the VTA).

Reviewer #2 (Remarks to the Author):

The manuscript explores the relationship between gene expression changes in the ventral tegmental area and dopamine dynamics in the nucleus accumbens in male rhesus macaques with an extensive alcohol drinking history interspersed with three forced abstinence periods. The measurements were done after the final abstinence period and thus the authors are examining changes that persist into abstinence, which is highly relevant to understanding persistent brain and behavior changes in alcohol use disorder (AUD). The authors hypothesized that this alcohol drinking history would induce a dysregulation in the expression of dopamine neuron-specific genes in the VTA. The authors found that in the full cohort (n=17) this alcohol drinking and abstinence history did not induce changes in gene expression (using full transcriptomic RNA-sequencing) in the VTA relative to the controls (n=11). In parallel to the gene expression analyses, 8 macaque drinkers and 8 controls were used to assess dopamine dynamics in the nucleus accumbens dopamine terminals using voltametric detection. However, it is not clear how these subjects were selected from the larger cohort. Here the authors found increased dopamine reuptake in the subjects with the alcohol drinking/abstinence history, but no change in dopamine release at the 350 uA stimulation (though decreased dopamine release was evoked by high intensity stimulations). Next, the authors examined the relation between gene expression in the VTA and accumbal dopamine dynamics using correlation analyses. The analysis extended to genes that encode for regulators of dopamine release such as the kappa opioid receptor (OPRK1) and others (DRD2, SLC6A3, SLC18A2). In the drinkers it was found that DRD2 and OPRK1 positively correlated with dopamine release in the nucleus accumbens. This correlation was not observed in the controls. This led to a focus on kappa opioid receptor which may be an important target for therapeutic development for the treatment of alcohol use disorder.

Comment 1.1: Overall, the focus on this extensive alcohol drinking and abstinence is important for the field. There are numerous strengths of this work including the replication of the literature showing a hypodopaminergic state following an extensive alcohol history, associations between gene expression and dopamine release, inclusion of pharmacological manipulation of kappa opioid receptor (KOR) on dopamine function in the nucleus accumbens to delve into the functional changes, and using a highly translational model.

Comment 1.2: My overall concern is that the major findings of this study are corollary. **1.3:** Alcohol drinking history did affect accumbal dopamine dynamics which is consistent with the literature, so that is good that it was confirmed, but this is not new information, and gene expression in the VTA was unchanged by drinking history. Therefore, it feels like a stretch to then be examining correlations between gene expression changes and accumbal dopamine. The idea that synchrony between functional release and upstream transcription is disrupted is not entirely compelling and feels like an overstatement. How are these changes functionally related? Other concerns are described below.

Response 1.1: We thank the reviewer for their time and constructive feedback, and for highlighting the importance of this work.

Response 1.2: We respectfully disagree that the major findings of the study are corollary. It is true that there are an uncommonly high number of scatter plots in the manuscript, because we wanted to show all of the data that led us through our line of reasoning, but the conclusions are almost entirely based on causal experimental designs and statistical tests. In fact, perhaps the most broadly impactful finding is the null effects observed in several thousand tests of causality on VTA gene expression between control and drinking subjects (see **Response 3.2** detailing why it is scientifically and statistically sound, and conceptually important that the null claims are considered as *prima facie* evidence).

Furthermore, the conclusions regarding changes in expression x function relationships are based on causal statistical tests (**Figure 3M, Figure 4M, Supplemental Figure 4I**). We do not make any claim of causal relationships based on correlation between a particular gene and function (if we imply this anywhere, please bring it to our attention); we do however claim that there is a causal change, induced by alcohol, in the probability that VTA gene expression covaries with the accumbal functional outputs that their encoded proteins are thought to contribute to. We feel that this claim carries significant implications, and as such felt that it was important to show every step of the analysis to demonstrate, for ourselves and the reader, that the rigor of the approach was commensurate with such a claim. Assumptions based on the idea that transcriptional expression is positively correlated with protein function are deeply intertwined with the foundations of neuroscience and biology in general – if the existence and directionality of these relationships can be experience-dependent, as

we claim here based on extensive analysis and empirical findings for dopaminergic and dopamine-regulating transcripts, there are major implications for the alcohol field and beyond.

As to whether the results are compelling, we have found them to be highly incongruent with our mental model of transcription but ultimately equally compelling. As detailed in Response 3, we did not come to these conclusions haphazardly nor were they what we hypothesized – the sequencing results were replicated across three independent, blinded analyses, and the voltammetry was collected and analyzed by separate experimenters blinded to the sequencing. Alcohol self-administration and behavioral analyses were conducted separate and blinded to both the functional and genetic experiments and analyses. Before revising our long-held hypothesis, we also revisited the literature in search of support for some of the assumptions that these data challenge, but have been unable to find analogous reports where synaptic function and endogenous gene expression (i.e. excluding knock-out/knock-down studies) were directly assessed and compared within-subject. Given the exhaustive analysis, well-powered sample size, sound statistical and experimental design, and in the absence, to our knowledge, of any direct refuting data we feel compelled to report these findings and consider the implications, however unexpected or incongruent with our assumptions. We have also, through discussion internally in the lab, reached the same conclusions across multiple interpretational frameworks and lines of reasoning. We are, of course, open to considering any alternative reasoning regarding our current claims of causality in the observed effects in drinking subjects versus controls.

Response 1.3: Regarding replication of prior literature in rodents, we are not aware of analogous reports examining time-resolved dopamine dynamics or genome wide transcriptional changes in dopamine neurons following chronic drinking and abstinence in rodents or non-human primates. While it is true that extensive work has been done in VTA and NAc of both model species, mostly following vapor exposure in rodents and drinking in macaques, these studies were performed in non-abstinent subjects and make no claims regarding persistent alterations that outlast the turnover of synaptic proteins controlling dopamine release and reuptake. The few rodent studies that have examined short-term withdrawal have not tested beyond 72 hours and have found that some of the vapor-induced changes in dopamine regulation dissipate within a few days (Karkhanis et al., 2015), and the KOR-specific effects have not been assessed. Here, we are measuring the function of receptors and transporters which, based on their biological half-life (Fleckenstein et al., 1996; Kimmel et al., 2003), have never interacted directly with alcohol; this is a critical distinction as understanding which alterations persist despite having never ‘seen’ alcohol and how this may occur is requisite for understanding the neurobiological basis of relapse.

Thus, the novel findings of the voltammetry experiments include **1)** that DAT and kappa opioid receptor plasticity persist into protracted abstinence despite not having interacted with alcohol directly, **2)** that changes in evoked dopamine release magnitude do not persist, except at the extremes of the input/output curve, **3)** that G protein signaling is both necessary and sufficient for kappa opioid receptor inhibitory control of axonal dopamine release, **4)** that G protein-mediated kappa opioid receptor signaling does not account for alcohol-induced supersensitivity (preliminary evidence is also presented implicating β -arrestin 2), **5)** that kappa opioid receptor co-activation is necessary for CRF receptor-mediated inhibition of dopamine release but not altered by drinking history, and **6)** that dynorphin release probability, measured using a novel, time-resolved readout, is increased in protracted abstinence.

These are the advances from the voltammetry experiments in isolation; as mentioned above and detailed in the responses below we feel that the most impactful conclusions arise from the unexpected but highly consistent observation of experience-dependent plasticity in the presence and directionality of covariance between transcriptional expression and protein function, observed across protein classes. Finally, all of the findings mentioned above, to our knowledge, constitute advances irrespective of the model species, but it is also important that we show these changes occur in a volitional drinking model on the time scale over which human AUD patients often relapse (Allen et al., 2018; Witkiewitz, 2005). We feel strongly (and thank the reviewer for acknowledging) that studies in non-human primate subjects have inherent value for our understanding of neurobiology and advancing human health, for which there is currently no comparable substitute (Mitchell et al., 2018).

Comment 2: The description of the results from Figure 1 set up the reader for the expectation that there was going to be a focus on individual differences. To this end, there is no description of why the gene expression analyses were conducted in all the subjects, but only 8 from each group were selected for the recordings. A description of the justification for which subjects were selected is necessary as is defining what their alcohol drinking levels were.

Response 2: Subjects/tissue are very limited, thus we carefully plan the number of slices dedicated to each experiment. These experiments spanned two cohorts of rhesus macaques with different voltammetry measures probed across cohorts. Gene expression was collected in all subjects, thus all 28 are included in the analysis of differentially expressed genes and weighted gene correlation network analyses. Gene expression was also correlated with drinking behavior in all of the alcohol drinking subjects ($n = 17$), though no significant correlations were found (**Supplemental Figure 6**). Voltammetry experiments were different between cohorts with kappa opioid receptor sensitivity being probed in Cohort 1, and GPCR downstream effector contribution was measured in Cohort 2. In addition to having differential measurements, Cohort 2 consisted of 3 control animals and 9 drinkers; thus, any correlation analyses would not be readily interpretable due to differential power between the groups (see **Reviewer 1 Response 5**). Therefore, all function \times gene correlations were conducted in Cohort 1. We agree that it was an oversight not to show the behavioral data by cohort and have thus added it here and within the manuscript (**Figure R6, Supplemental Figure 1**). We have also expanded our description of the cohorts and rationale for the analyses in the methods section.

Comment 3.1: There were no changes in gene expression following a history of alcohol drinking/abstinence in the full cohort. However, were there any changes in the smaller subgroup that was used in the majority of the paper? **3.2:** Relatedly, it is conceptually confusing that drinking history doesn't affect gene expression changes but yet, the focus of the paper is on correlations between gene expression and dopamine dynamics.

Response 3.1: See **Response 2** for a detailed description of the cohort breakdowns. There were no changes in expression of the *a priori* selected genes in the subgroup used for the correlations (see **Supplemental Figure 5**) or when examined with all animals included (**Figure 2**).

Response 3.2: We agree with the reviewer that the lack of effect is quite surprising. Before addressing conceptual basis and interpretation, we would like to first clarify some of the steps taken to ensure that the results were in fact null. We hypothesized (and believe that much of the field also would) that **1**) chronic alcohol consumption and abstinence would result in robust changes in VTA gene expression, and **2**) that any long-lasting changes in dopamine terminal function (i.e., longer than the lifetime of the synaptic receptor or

transporter involved) would likely be related to changes in the expression of dopamine-specific genes. We found zero differentially expressed genes between drinkers and controls – this was highly surprising to us, and we spent two years after sequencing was completed verifying the findings before we were convinced of their veracity. This included three separate analyses of the dataset from independent groups using different alignment methods and differential expression pipelines (USCF Sequencing Core, Vanderbilt Data Solutions Core, Kiraly Lab at Wake Forest University). All three research groups were initially blinded to subject group assignment and voltammetry data – we did not include all three replications in the manuscript because they ultimately produced identical results: zero differentially expressed genes. The full dataset is posted on GEO (GEO accession: GSE244557) and we would be happy to provide these replication analyses directly if it would be helpful. It is also notable that the sequencing was performed with deeper read depth than is typical in the field (paired-end 150bp, roughly 150 million read pairs per sample) and with a more well-powered sample size than is typical in mouse or monkey work (11 controls, 17 drinkers). Both of these conditions would be expected to reduce type I and type II error, effects which are typically compounding/exponential when performing multiple comparisons tests, and appropriately adjusting for familywise error rates, across thousands of tests as is necessary for genome-wide assessment of differential expression.

While there are many papers that speak broadly in discussions and reviews about potential long-term alcohol induced changes in VTA transcription (including prior work from the authors), we are not aware of any VTA sequencing studies performed after long-term abstinence from alcohol self-administration in animal models. Thus, while we were surprised by the result conceptually, it is not in conflict with prior data. In terms of the explanation, we have expanded our discussion of the possible mechanisms, but the bottom line is, to be blunt, that we (the authors and the field) don't yet know the precise mechanisms by which alcohol-induced plasticity persists long-term. Our only point of contention is that this is not a shortcoming of the current work – many standing theories and ongoing experiments, throughout the alcohol field, rest on similar assumptions to those which we based our hypotheses on, which means a true null result is much more likely to impact the course of the field than a true positive result. The null result here, with what we feel is a sufficiently rigorous design and execution, is critical as it moves the needle from there being an absence of evidence for long-term alcohol-induced changes in VTA gene expression, to direct evidence of absence.

Regarding the focus on the correlations between function and expression, the initial selection of genes and functional measures for correlation were planned *a priori*, before we had reason to revise our hypothesis that chronic drinking would induce robust changes in VTA gene expression. The focus shifted after we performed the initial correlations and started to plot them next to each other – we were not looking for it, but it was difficult to ignore the patterns that emerged across the correlations (see **Figure 3E-L**). The causality of alcohol exposure in producing these patterns was tested for directly using non-correlative analyses: the null hypothesis was rejected in drinkers, but not controls, when analyzed in parallel with a one-sample design against a hypothetical value 0, as well as with a standard between-group hypothesis test statistic comparing mean slopes in drinkers versus controls.

The additional correlational analyses (i.e., rank ordered combined analysis and sample size-, protein class-matched control gene analysis) were planned *a posteriori* and were designed to determine whether this surprising positive effect held up to scrutiny in intersectional analyses. Importantly, these analyses were still planned and performed in a forward fashion; once control genes were selected, the analysis was performed identically to the *a priori* gene set and was not revisited. In regard to integrating these findings with the field, the fact that drinking and abstinence induced a causal change in the probability of covariance between gene expression and functional readouts mean that the data as a whole do not argue against the importance of VTA transcriptional regulation in mediating alcohol's lasting effects throughout abstinence; rather, that transcriptional involvement is not apparent in expression data *per se*. Most importantly, these results strongly suggest that basic assumptions regarding the relationship between gene expression and functional synaptic measures need to be empirically tested, rather than being entered into the literature as *de facto* truths via inductive reasoning in conceptual frameworks. We feel that these conclusions raise the importance of the findings to be relevant and of interest to the neuroscience field at large.

Comment 4: The functional studies of this manuscript focused on the effect of exogenous activation of the kappa opioid receptor system on dopamine release dynamics in the nucleus accumbens. Again, results

confirmed previous work. It will be important to highlight what aspect of these studies and the results that are novel.

Response 4: See **Response 1** for a more detailed description on the novelty of this work (we have also added these rationales throughout the results and discussion sections). Regarding augmented sensitivity of the kappa opioid receptor, to our knowledge this is the first functional measurement of this system during prolonged abstinence despite being consistently referred to as a cause of negative affective states during abstinence. Additionally, we show augmented endogenous dynorphin release and resulting inhibition of dopamine release, which to our knowledge has not been shown in any model species or alcohol exposure timepoint. This is the first demonstration of alcohol-induced plasticity in dynorphin release probability, using a functional, time-resolved measurement, in any preparation, in addition to being unique and important to the field given the highly translationally relevant model used here.

Comment 5: This study was conducted in male subjects. While I understand the limitations, given the review recommendations of the journal to consider sex in the area of study, it will be important to address this in the manuscript.

Response 5: We agree that ultimately it is critical to test these questions in both sexes. Due to the scarcity of the model species and the length of the experimental paradigm (the first cohort in this study began experiments in 2012), we perform the experiments as tissue becomes available. We and others have recently called for greater emphasis on sex x environment interactions rather than claims of sexual dimorphism when interpreting drug and alcohol self-administration data (Radke et al., 2021; Siciliano, 2019; Zachry et al., 2019); accordingly, we find that investigating within each sex in parallel, rather than attempts to parse sex differences *per se*, often produces more clear results when discovery of mechanistic reasons for dimorphisms is not the explicit goal (Brown et al., 2023). We have now added a discussion point regarding the need for similar experiments in female subjects. We hope that this is not perceived as an attempt to sidestep the issue – we sincerely plan to continue to pursue parallel investigations in both sexes, and have a track record of doing so in prior work in rodent and monkey models (Brown et al., 2023; Nolan et al., 2023; Siciliano et al., 2015, 2016, 2019).

Though we plan to pursue experiments in female subjects, it is also worth noting that the NIH has exempted non-human primates from the mandate to study both sexes, in part because there is a shortage of rhesus monkeys available for research (Nonhuman Primate Evaluation and Analyses, final report. Office of Infrastructure Programs, NIH, July 12, 2024). The only way to increase production is to sequester greater numbers of female rhesus as breeders, placing even more pressure on enrolling female monkeys into research protocols.

Comment 6: I would recommend that the authors clarify that the gene expression data is from the VTA and the dopamine measures are from the accumbens – clarifying this especially in the figures would be helpful. This also applies to statements of hypotheses and conclusions. It is not that alcohol-induced changes in gene expression were not observed, but rather they were not observed in the VTA.

Response 6: We agree with the reviewer completely, and did not intend to make a more general claim regarding a broad lack of effect of alcohol on gene expression; we have thus reiterated that the gene expression was measured in the VTA within the text, figure legends, axes titles, and have modified the experimental diagram schematic in Figure 1 to further illustrate the analysis method used throughout. We have taken care to ensure that this is accurately articulated throughout but welcome any additional feedback on this issue.

We have also expanded our discussion of the data in the results and discussion section to highlight the importance of the work given the surprising finding, as well as the statistical rigor of the design of the gene expression comparisons (see **Response 3.2**).

Reviewer #3 (Remarks to the Author):

Comment 1: I co-reviewed this manuscript with one of the reviewers who provided the listed reports. This is part of the Nature Communications initiative to facilitate training in peer review and to provide appropriate recognition for Early Career Researchers who co-review manuscripts

Response 1: We appreciate this reviewer taking the time to contribute to one of the listed reports and are grateful that our manuscript was able to be a part of a worthy training initiative.

References

- Allen, D. C., Gonzales, S. W., & Grant, K. A. (2018). Effect of repeated abstinence on chronic ethanol self-administration in the rhesus monkey. *Psychopharmacology*, *235*(1), 109–120.
<https://doi.org/10.1007/s00213-017-4748-9>
- Bazov, I., Sarkisyan, D., Kononenko, O., Watanabe, H., Yakovleva, T., Hansson, A. C., Sommer, W. H., Spanagel, R., & Bakalkin, G. (2018). Dynorphin and κ -Opioid Receptor Dysregulation in the Dopaminergic Reward System of Human Alcoholics. *Molecular Neurobiology*, *55*(8), 7049–7061.
<https://doi.org/10.1007/s12035-017-0844-4>
- Brown, A. R., Branthwaite, H. E., Farahbakhsh, Z. Z., Mukerjee, S., Melugin, P. R., Song, K., Noamany, H., & Siciliano, C. A. (2023). Structured tracking of alcohol reinforcement (STAR) for basic and translational alcohol research. *Molecular Psychiatry*, *28*(4), 1585–1598. <https://doi.org/10.1038/s41380-023-01994-4>
- De Santis, S., Cosa-Linan, A., Garcia-Hernandez, R., Dmytrenko, L., Vargova, L., Vorisek, I., Stopponi, S., Bach, P., Kirsch, P., Kiefer, F., Ciccocioppo, R., Sykova, E., Moratal, D., Sommer, W. H., & Canals, S. (2020). Chronic alcohol consumption alters extracellular space geometry and transmitter diffusion in the brain. *Science Advances*, *6*(26), eaba0154. <https://doi.org/10.1126/sciadv.aba0154>
- Fleckenstein, A. E., Pögün, S., Carroll, F. I., & Kuhar, M. J. (1996). Recovery of dopamine transporter binding and function after intrastriatal administration of the irreversible inhibitor RTI-76 [3 beta-(3p-chlorophenyl) tropan-2 beta-carboxylic acid p-isothiocyanatophenylethyl ester hydrochloride]. *The Journal of Pharmacology and Experimental Therapeutics*, *279*(1), 200–206.
- Janse, R. J., Hoekstra, T., Jager, K. J., Zoccali, C., Tripepi, G., Dekker, F. W., & van Diepen, M. (2021). Conducting correlation analysis: Important limitations and pitfalls. *Clinical Kidney Journal*, *14*(11), 2332–2337. <https://doi.org/10.1093/ckj/sfab085>
- Karkhanis, A. N., Rose, J. H., Huggins, K. N., Konstantopoulos, J. K., & Jones, S. R. (2015). Chronic intermittent ethanol exposure reduces presynaptic dopamine neurotransmission in the mouse nucleus accumbens. *Drug and Alcohol Dependence*, *150*, 24–30.
<https://doi.org/10.1016/j.drugalcdep.2015.01.019>

- Kimmel, H. L., Carroll, F. I., & Kuhar, M. J. (2003). Withdrawal from repeated cocaine alters dopamine transporter protein turnover in the rat striatum. *The Journal of Pharmacology and Experimental Therapeutics*, *304*(1), 15–21. <https://doi.org/10.1124/jpet.102.038018>
- Kissler, J. L., Sirohi, S., Reis, D. J., Jansen, H. T., Quock, R. M., Smith, D. G., & Walker, B. M. (2014). The one-two punch of alcoholism: Role of central amygdala dynorphins/kappa-opioid receptors. *Biological Psychiatry*. <https://doi.org/10.1016/j.biopsych.2013.03.014>
- Love, M. I., Huber, W., & Anders, S. (2014). Moderated estimation of fold change and dispersion for RNA-seq data with DESeq2. *Genome Biology*, *15*(12), 550. <https://doi.org/10.1186/s13059-014-0550-8>
- Mitchell, A. S., Thiele, A., Petkov, C. I., Roberts, A., Robbins, T. W., Schultz, W., & Lemon, R. (2018). Continued need for non-human primate neuroscience research. *Current Biology*, *28*(20), R1186–R1187. <https://doi.org/10.1016/j.cub.2018.09.029>
- Nolan, S. O., Melugin, P. R., Erickson, K. R., Adams, W. R., Farahbakhsh, Z. Z., Mcgonigle, C. E., Kwon, M. H., Costa, V. D., Lapish, C. C., Hackett, T. A., Carlson, V. C. C., Constantinidis, C., Grant, K. A., & Siciliano, C. A. (2023). *Recurrent activity within microcircuits of macaque dorsolateral prefrontal cortex tracks cognitive flexibility* (p. 2023.09.23.559125). bioRxiv. <https://doi.org/10.1101/2023.09.23.559125>
- Pearson, K. (1896). VII. Mathematical contributions to the theory of evolution.—III. Regression, heredity, and panmixia. *Philosophical Transactions of the Royal Society of London. Series A, Containing Papers of a Mathematical or Physical Character*, *187*, 253–318. <https://doi.org/10.1098/rsta.1896.0007>
- Phillips, R. A., Tuscher, J. J., Black, S. L., Andraka, E., Fitzgerald, N. D., Ianov, L., & Day, J. J. (2022). An atlas of transcriptionally defined cell populations in the rat ventral tegmental area. *Cell Reports*, *39*(1). <https://doi.org/10.1016/j.celrep.2022.110616>
- Radke, A. K., Sneddon, E. A., Frasier, R. M., & Hopf, F. W. (2021). Recent Perspectives on Sex Differences in Compulsion-Like and Binge Alcohol Drinking. *International Journal of Molecular Sciences*, *22*(7), 3788. <https://doi.org/10.3390/ijms22073788>
- Rothblat, D. S., Rubin, E., & Schneider, J. S. (2001). Effects of chronic alcohol ingestion on the mesostriatal dopamine system in the rat. *Neuroscience Letters*, *300*(2), 63–66. [https://doi.org/10.1016/S0304-3940\(01\)01548-8](https://doi.org/10.1016/S0304-3940(01)01548-8)

- Siciliano, C. A. (2019). Capturing the complexity of sex differences requires multidimensional behavioral models. *Neuropsychopharmacology*, *44*(12), 1997–1998. <https://doi.org/10.1038/s41386-019-0424-6>
- Siciliano, C. A., Calipari, E. S., Cuzon Carlson, V. C., Helms, C. M., Lovinger, D. M., Grant, K. A., & Jones, S. R. (2015). Voluntary ethanol intake predicts κ -opioid receptor supersensitivity and regionally distinct dopaminergic adaptations in macaques. *Journal of Neuroscience*, *35*(15), 5959–5968. <https://doi.org/10.1523/JNEUROSCI.4820-14.2015>
- Siciliano, C. A., Calipari, E. S., Yorgason, J. T., Lovinger, D. M., Mateo, Y., Jimenez, V. A., Helms, C. M., Grant, K. A., & Jones, S. R. (2016). Increased presynaptic regulation of dopamine neurotransmission in the nucleus accumbens core following chronic ethanol self-administration in female macaques. *Psychopharmacology*, *233*(8), 1435–1443. <https://doi.org/10.1007/s00213-016-4239-4>
- Siciliano, C. A., Karkhanis, A. N., Holleran, K. M., Melchior, J. R., & Jones, S. R. (2018). Cross-Species Alterations in Synaptic Dopamine Regulation After Chronic Alcohol Exposure. *Handbook of Experimental Pharmacology*, *248*, 213–238. https://doi.org/10.1007/164_2018_106
- Siciliano, C. A., Mauterer, M. I., Fordahl, S. C., & Jones, S. R. (2019). Modulation of striatal dopamine dynamics by cocaine self-administration and amphetamine treatment in female rats. *The European Journal of Neuroscience*, *50*(4), 2740–2749. <https://doi.org/10.1111/ejn.14437>
- Simpson, E. H. (1951). The Interpretation of Interaction in Contingency Tables. *Journal of the Royal Statistical Society: Series B (Methodological)*, *13*(2), 238–241. <https://doi.org/10.1111/j.2517-6161.1951.tb00088.x>
- Swanson, L. W. (1982). The projections of the ventral tegmental area and adjacent regions: A combined fluorescent retrograde tracer and immunofluorescence study in the rat. *Brain Research Bulletin*, *9*(1–6), 321–353. [https://doi.org/10.1016/0361-9230\(82\)90145-9](https://doi.org/10.1016/0361-9230(82)90145-9)
- Witkiewitz, K. (2005). Defining Relapse from a Harm Reduction Perspective. *Journal of Evidence-Based Social Work*, *2*(1–2), 191–206. https://doi.org/10.1300/J394v02n01_11
- Yule, G. U. (1903). Notes on the Theory of Association of Attributes in Statistics. *Biometrika*, *2*(2), 121–134. <https://doi.org/10.2307/2331677>

- Zachry, J. E., Johnson, A. R., & Calipari, E. S. (2019). Sex Differences in Value-Based Decision Making Underlie Substance Use Disorders in Females. *Alcohol and Alcoholism (Oxford, Oxfordshire)*, 54(4), 339–341. <https://doi.org/10.1093/alcalc/agz052>
- Zhou, F. C., Zhang, J. K., Lumeng, L., & Li, T.-K. (1995). Mesolimbic dopamine system in alcohol-preferring rats. *Alcohol*, 12(5), 403–412. [https://doi.org/10.1016/0741-8329\(95\)00010-O](https://doi.org/10.1016/0741-8329(95)00010-O)

We would like to thank the reviewers and editors for their time and constructive comments on our manuscript.

Responses to Individual Reviewer Comments:

Please find point-by-point responses to each reviewer's comments, colored coded by reviewer:

Responses to Reviewer #1

Responses to Reviewer #2

A clean copy of the revised manuscript is provided.

Reviewer #1 (Remarks to the Author):

The revisions include clarifications and additional analyses, which are mainly included in the rebuttal letter, not the manuscript. Additionally, the authors have only partially addressed my comments. Here are a few examples:

We apologize for misunderstanding some of the prior comments, and have done our best to re-address them below.

Comment 2: The authors addressed the second part of the question but did not explain why the voltammetry input-output curve is not shown in the main figure. To be more direct, I request that the authors display the voltammetry input-output curve in Figure 3.

Apologies for misunderstanding the intent of the comment. Given the size of (previous) Figure 3, we felt that adding the input-output panels into that figure would be overwhelming. Thus, we have now included the input-output panels as their own main figure (current Figure 3) in the manuscript. We would be happy to adjust its placement if this change is not acceptable.

Comment 3: My question was: "Are the changes in max DA and reuptake correlated to each other?"

The authors responded with a correlation using DA amplitudes that are not maximal, and they used EA50 instead.

- a) This plot does not answer my comment.
- b) The plot shows a positive correlation between reuptake speed and dopamine release, which contradicts the findings of the study. The results show faster reuptake and lower max release.
- c) I assume the EA50 is the same intensity for both groups. Could the authors please confirm this? Otherwise, these are not direct comparisons.

- a) We apologize for misreading the prior comment, we missed the 'max' and thought you were inquiring about reuptake vs release in general. We have now included the correlation between dopamine evoked by the highest stimulation intensity tested and V_{\max} (**Figure R1**). As the input/output curve was conducted on a cohort with only three controls, we did not run a correlation on that group. In drinkers, V_{\max} does not correlate with the maximal dopamine release observed.

- b) As noted by the reviewer, the prior correlation was with the EA_{50} values, which were not changed; we only saw a decrease in release at the highest stimulation intensities. As shown above, when correlated with max stimulation intensity, which probes the size of the releasable pool as opposed to release probability, we no longer see the correlation between release and reuptake.
- c) Correct, identical stimulation parameters were used between controls and drinkers in all cases. The EA_{50} were nearly identical (controls: $EA_{50} = 316.4\mu\text{A}$; drinkers: $EA_{50} = 315.9\mu\text{A}$), and were rounded to $350\mu\text{A}$ when used for subsequent experiments and this comparison was shown in what is now Figure 3.

There are also other contradictions throughout the manuscript that are hard to reconcile. For example, the results section states: "However, the maximal rate of dopamine reuptake (V_{max}) was increased in subjects with a history of chronic alcohol self-administration (Figure 3C), suggesting an upregulation of dopamine transporter activity and faster removal of dopamine from the extracellular space, thereby decreasing tonic extracellular dopamine levels."

However, the study does not measure tonic dopamine levels, but rather evoked dopamine release. This discrepancy needs to be clarified.

Additionally, in the rebuttal letter, the authors use an opposing argument to address comment 3: "Essentially, when uptake rate is high, there is less dopamine outside and thus more dopamine inside the terminal, which leads to high release as well."

This contradiction between the results section and the rebuttal response requires resolution.

We did not intend to claim that we were showing decreased tonic extracellular levels. We were suggesting this is a likely consequence of increased reuptake, as has been shown in prior work (Ferris et al., 2014). We have reworded this passage to ensure that data reporting and speculating are unambiguous:

"However, maximal rate of dopamine reuptake (V_{max}) was increased in subjects with a history of chronic alcohol self-administration (Figure 4C). These results are consistent with a hypodopaminergic state⁴⁶⁻⁴⁸, which is consistently observed in the mesolimbic system of humans with AUD and thought to be critical to symptomatology⁴⁹⁻⁵³, and provide evidence that this state persists in prolonged abstinence."

Regarding the relationship between evoked dopamine release (as measured in the manuscript) and tonic extracellular dopamine levels *in vivo* (speculated but not measured), these measures are dissociable. Evoked release assays the readily releasable pool and terminal excitability/release probability, depending on stimulation intensity, as discussed above. Tonic extracellular levels, typically measured with microdialysis, are influenced by many factors including somatic activity of dopamine neurons, local modulation of terminals by acetylcholine and other retrograde signaling mechanisms rate of metabolism, extracellular tortuosity, etc. Of these many mechanisms, reuptake of dopamine through the dopamine transporter has been shown to be most influential (Ferris et al., 2014; Jones et al., 1998). We have now ensured that any references to tonic dopamine levels are clearly disambiguated from evoked release throughout the manuscript.

Comment 4: The response could be improved. The authors argue that while the suggestion is valid, they have already used the available brain tissue. However, they have access to either fixed or fresh tissue from alcohol drinkers, which could be used for immunostaining or western blot analysis. I recommend considering this option.

We apologize for the confusion – we were interpreting the requested experiment to be specifically regarding the expression \times function correlations referenced in the comment; to further explore this finding would require tissue that was collected from animals that had also undergone voltammetry in the accumbens and/or sequencing of the ventral tegmental area. Regardless, to our knowledge, these two cohorts are the only ones run to date where necropsy was performed at a 30-day withdrawal period – other tissue available to us is entirely from cohorts that were sacrificed at a 0-withdrawal time point (i.e., alcohol access until the morning of necropsy). Thus, while we do have access to some alcohol-exposed non-human primate tissue, none of the available tissue to our knowledge would allow for correlations with the other measurements nor would be appropriate for indirect comparison given the different paradigms used.

Comment 7-8: I appreciate the additional analysis for correlation of DYN and kappa receptor expression, as well as the clarification on the tissue collection. A remaining question is: How do authors interpret the negative correlation between kappa receptor expression in VTA and inhibition of DA release in the Nac, the fact that more inhibition is seen when the receptor expression is lower? This is counterintuitive.

We agree that the negative correlation between kappa opioid receptor RNA expression in the VTA and kappa opioid receptor mediated inhibition of dopamine release in the NAc is initially counterintuitive, but through this work, we realized that it is only surprising in the context of current assumptions the field holds regarding the relationship between transcript expression and protein function. The assumption that transcriptional expression is positively correlated with protein function is deeply intertwined with the foundations of neuroscience and biology in general. However, in a search of the literature for evidence supporting this assumption, we were unable to find analogous reports where synaptic function and endogenous gene expression (i.e. excluding knock-out/knock-down studies) were directly assessed and compared within-subject. Not only that, but there have also been reports that transcript expression is minimally predictive of protein levels and sometimes even inversely correlated (Franks et al., 2017; Gonçalves et al., 2017; Moritz et al., 2019; Wei et al., 2015). In the revised manuscript discussion, we speculate a number of ways this relationship can be modulated (or reversed from our expectations in the case of Oprk1):

“Experience-dependent shifts in a plethora of biological processes such as alterations in rate of protein translation, post-translational modifications, and trafficking could potentially explain these results. For example, a higher level of DNA transcription may not be reflected in RNA read counts if the RNA is being translated to protein at a faster rate, resulting in a decorrelation between RNA counts and protein expression. This could occur through a range of mechanisms including rate of 5' capping, mRNA stability, or localization. Even in cases where a given gene is primarily regulated at the level of transcription and RNA levels are correlated with protein expression, if post-transcriptional alterations, such as RNA editing or alternative splicing, dictates protein activity, transcript expression \times function relationships may not be present. Likewise, any number of post-translational modifications could alter functional outputs independent of transcript expression, through differential trafficking, affinity states, and rates of protein degradation. These examples only constitute a small number of possible explanations.”

Reviewer #2 (Remarks to the Author):

I really appreciate the care with which the authors responded to my comments and the thorough explanations. The clarifications/additions in the text are also very helpful. My concerns were all addressed and I have no additional comments/concerns.

Thank you for the kind words and the comments which helped us improve this manuscript.

Reviewer #3 (Remarks to the Author):

- Ferris, M. J., España, R. A., Locke, J. L., Konstantopoulos, J. K., Rose, J. H., Chen, R., & Jones, S. R. (2014). Dopamine transporters govern diurnal variation in extracellular dopamine tone. *Proceedings of the National Academy of Sciences of the United States of America*, *111*(26), E2751–E2759. <https://doi.org/10.1073/pnas.1407935111>
- Franks, A., Airoidi, E., & Slavov, N. (2017). Post-transcriptional regulation across human tissues. *PLOS Computational Biology*, *13*(5), e1005535. <https://doi.org/10.1371/journal.pcbi.1005535>
- Gonçalves, E., Fragoulis, A., Garcia-Alonso, L., Cramer, T., Saez-Rodriguez, J., & Beltrao, P. (2017). Widespread Post-transcriptional Attenuation of Genomic Copy-Number Variation in Cancer. *Cell Systems*, *5*(4), 386-398.e4. <https://doi.org/10.1016/j.cels.2017.08.013>
- Jones, S. R., Gainetdinov, R. R., Jaber, M., Giros, B., Wightman, R. M., & Caron, M. G. (1998). Profound neuronal plasticity in response to inactivation of the dopamine transporter. *Proceedings of the National Academy of Sciences*, *95*(7), 4029–4034. <https://doi.org/10.1073/pnas.95.7.4029>
- Moritz, C. P., Mühlhaus, T., Tenzer, S., Schulenburg, T., & Friauf, E. (2019). Poor transcript-protein correlation in the brain: Negatively correlating gene products reveal neuronal polarity as a potential cause. *Journal of Neurochemistry*, *149*(5), 582–604. <https://doi.org/10.1111/jnc.14664>
- Wei, Y.-N., Hu, H.-Y., Xie, G.-C., Fu, N., Ning, Z.-B., Zeng, R., & Khaitovich, P. (2015). Transcript and protein expression decoupling reveals RNA binding proteins and miRNAs as potential modulators of human aging. *Genome Biology*, *16*(1), 41. <https://doi.org/10.1186/s13059-015-0608-2>